# FLOW MATCHING WITH INJECTED NOISE FOR OFFLINE-TO-ONLINE REINFORCEMENT LEARNING

**Yongjae Shin**  **Jongseong Chae**  **Jongeui Park**  **Youngchul Sung** [*]
KAIST
{yongjae.shin, ycsung}@kaist.ac.kr

## ABSTRACT

Generative models have recently demonstrated remarkable success across diverse domains, motivating their adoption as expressive policies in reinforcement learning (RL). While they have shown strong performance in offline RL, particularly where the target distribution is well defined, their extension to online fine-tuning has largely been treated as a direct continuation of offline pre-training, leaving key challenges unaddressed. In this paper, we propose Flow Matching with Injected Noise for Offline-to-Online RL (FINO), a novel method that leverages flow matching-based policies to enhance sample efficiency for offline-to-online RL. FINO facilitates effective exploration by injecting noise into policy training, thereby encouraging a broader range of actions beyond those observed in the offline dataset. In addition to exploration-enhanced flow policy training, we combine an entropy-guided sampling mechanism to balance exploration and exploitation, allowing the policy to adapt its behavior throughout online fine-tuning. Experiments across diverse, challenging tasks demonstrate that FINO consistently achieves superior performance under limited online budgets.

## 1 INTRODUCTION

Generative models have recently demonstrated substantial success across diverse domains, producing high-quality outputs in areas such as text and image (Brown et al., 2020; Rombach et al., 2022). By leveraging their expressive capacity, these models can capture complex or multimodal distributions present in the underlying datasets, beyond the reach of conventional parametric models. This opens up new opportunities in reinforcement learning (RL), particularly for policy design.

Since a policy in RL can be regarded as a generative model conditioned on states, there has been increasing interest in applying generative modeling to policy design, such as denoising diffusion (Sohl-Dickstein et al., 2015; Ho et al., 2020) and flow matching (Lipman et al., 2023; Albergo & Vanden-Eijnden, 2023). While Gaussian policies have been the conventional choice, they often struggle to represent multimodal or high-dimensional action distributions (Park et al., 2024). By contrast, generative policies provide the expressivity required to handle complex RL tasks and have demonstrated superior performance, particularly in offline RL where the target distribution is explicitly defined (Wang et al., 2023b; Hansen-Estruch et al., 2023; Kang et al., 2023; Zhang et al., 2025; Fang et al., 2025; Park et al., 2025b).

Despite such expressivity, offline RL inherently suffers from a fundamental limitation in that the performance of the policy is constrained by the quality of the offline dataset. Accordingly, offline-to-online RL has been proposed to address this issue, enabling a pre-trained policy to further improve its performance through short-term direct interaction with the environment (Lee et al., 2022; Zhang et al., 2023; Nakamoto et al., 2023; Zhang et al., 2024; Zhou et al., 2024). While some studies (Hansen-Estruch et al., 2023; Park et al., 2025b) have incorporated additional online fine-tuning of generative policies, they merely treat it as a continuation of offline pre-training rather than designing approaches specialized to the online fine-tuning.

As offline-to-online RL encompasses both offline and online stages, it naturally introduces challenges beyond those faced in a purely offline RL framework. Unlike offline RL, which relies solely

---

[*]Corresponding author.    Our code is available at `https://github.com/CTID282/FINO`.

on pre-collected datasets, offline-to-online RL incorporates an online fine-tuning phase, making it beneficial to design the offline pre-training with this subsequent stage in mind from the beginning. At the same time, the framework raises the practical question of how to best exploit the pre-trained policy during online fine-tuning. Thus, framing offline-to-online RL merely as an extension of offline RL can limit the extent to which its potential is realized.

In this work, we propose Flow Matching with Injected Noise for Offline-to-Online RL (FINO), a novel policy learning approach for the offline-to-online RL framework. Motivated by recent findings that maintaining diversity facilitates more effective fine-tuning (Fan et al., 2025; Li et al., 2025; Zhai et al., 2025; Sorokin et al., 2025), we introduce a training strategy that injects noise into the flow matching to explicitly promote diversity in the policy from the beginning of offline pre-training. This injection encourages the policy to learn a broader range of action space than that present in the offline dataset, thereby establishing a strong foundation for exploration during online fine-tuning. To effectively leverage this during online fine-tuning, we introduce an entropy-guided sampling mechanism that exploits the acquired diversity for exploration while balancing exploration and exploitation by adapting to the evolving behavior of the policy. We experiment on 45 diverse and challenging tasks from OGBench (Park et al., 2025a) and D4RL (Fu et al., 2020) under a limited online fine-tuning budget. The results show that FINO achieves consistently strong performance across tasks, even in complex environments, thereby demonstrating FINO as an effective and reliable approach for offline-to-online RL.

## 2 PRELIMINARIES

**Offline-to-Online Reinforcement Learning.** In this paper, we consider a Markov Decision Process (MDP) (Sutton et al., 1998) $\mathcal{M} = (\mathcal{S}, \mathcal{A}, r, \mathcal{P}, \gamma)$, where $\mathcal{S}$ denotes the state space, $\mathcal{A}$ the action space, $r$ the reward function, $\mathcal{P}$ the transition probability distribution, and $\gamma$ the discount factor. The objective of RL is to train a policy that maximizes the expected cumulative return $\mathbb{E}_\pi[\sum_i \gamma^i r(s_i, a_i)]$. Offline-to-online RL is a two-stage learning framework consisting of offline pre-training and online fine-tuning (Lee et al., 2022; Zhang et al., 2023; Nakamoto et al., 2023; Zhang et al., 2024; Zhou et al., 2024). This framework is designed to combine the strengths of offline and online RL: the stability gained from pre-collected datasets without interactions and the adaptability that comes from environment interaction. In the offline pre-training, a policy is trained on a static dataset $D = \{(s, a, r, s')\}$, providing a reliable initialization. Subsequently, during the online fine-tuning, the pre-trained policy directly interacts with the environment, allowing it to refine its behavior and correct limitations inherited from the offline dataset.

**Flow Matching.** Flow matching (Lipman et al., 2023; Albergo & Vanden-Eijnden, 2023) is a generative modeling framework that constructs a transformation between two probability distributions via ordinary differential equations (ODEs). Unlike diffusion models (Sohl-Dickstein et al., 2015; Ho et al., 2020), which rely on stochastic differential equations (SDEs), flow matching is based on deterministic ODEs. This design not only simplifies training but also enables faster inference.

The central component of flow matching is a time-dependent vector field $v_\theta(t, x)$ that defines a flow $\phi_t$, mapping a base distribution $p_0$ into a target data distribution $p_1$:

$$\frac{d}{dt}\phi_t(x) = v_\theta(t, \phi_t(x)), \quad \phi_0(x) = x. \tag{1}$$

A widely used formulation of flow matching is based on Optimal Transport (OT) (Lipman et al., 2023), where transformations are constructed by linearly interpolating between samples from the base and target distributions ($x_0 \sim p_0$ and $x_1 \sim p_1$), with the interpolation time $t$ sampled uniformly:

$$x_t = (1 - t)x_0 + tx_1, \quad t \sim \text{Unif}([0, 1]). \tag{2}$$

The vector field is trained to align its prediction with the direction of this linear path:

$$\min_\theta \mathbb{E}_{\substack{x_0 \sim p_0, x_1 \sim p_1, \\ t \sim \text{Unif}([0,1])}} \left[ ||v_\theta(t, x_t) - (x_1 - (1 - \sigma_{\min})x_0)||_2^2 \right]. \tag{3}$$

where $\sigma_{\min}$ is a sufficiently small constant. Once the vector field $v_\theta$ is trained, generation is performed by sampling $x_0 \sim p_0$ and solving the learned ODE until $t = 1$ to obtain $\phi_1(x_0) \sim p_1$. In this work, we use the Euler method to solve the ODE for sample generation.

**Flow Q-Learning.** Flow Q-Learning (FQL) (Park et al., 2025b) applies flow matching to policy design for offline RL. It formulates the policy as a state-conditioned flow model and trains it by adapting flow matching to behavior cloning:

$$\mathcal{L}_\pi(\theta) = \mathbb{E}_{\substack{x_0 \sim \mathcal{N}(0,I), \\ s,a=x_1 \sim D, \\ t \sim \mathrm{Unif}([0,1])}} \left[ ||v_\theta(t,s,x_t) - (x_1 - x_0)||_2^2 \right]. \tag{4}$$

Integrating the trained vector field $v_\theta$ induces a mapping $a_\theta(s,z)$ from state $s$ and noise $z$ to action, which defines a policy $\beta_\theta$, linking the flow formulation to a policy representation.

To enable efficient training, FQL further introduces a one-step policy $\pi_\omega$, which is jointly optimized by distillation from the flow policy and action-value maximization:

$$\mathcal{L}_\pi(\omega) = \mathbb{E}_{\substack{s \sim D, \\ z \sim \mathcal{N}(0,I), \\ a_\omega(s,z) \sim \pi_\omega}} \left[ -Q_\phi(s, a_\omega(s,z)) + \alpha \|a_\omega(s,z) - a_\theta(s,z)\|_2^2 \right], \tag{5}$$

where $\alpha$ is a hyperparameter. In practice, the one-step policy provides a direct mapping from noise to actions without sequential ODE integration, enabling efficient action selection while inheriting the expressiveness of the flow model.

## 3 MOTIVATION

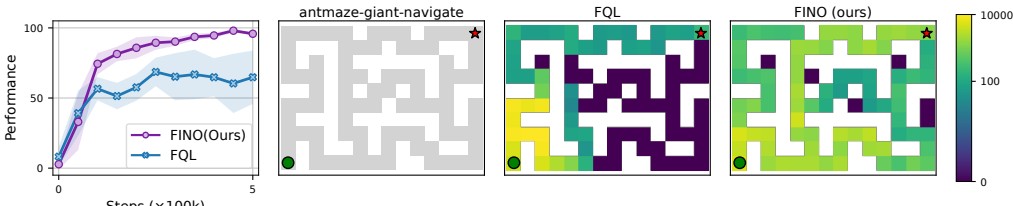

Figure 1: Comparison of FQL and FINO (ours) in terms of performance and exploration patterns on the environment `antmaze-giant-navigate`. The green circle and red star indicate the initial and goal states, respectively.

Our motivation lies in better leveraging the expressivity of generative policy, flow policy in particular, to address the challenges of offline-to-online RL. There exist prior studies in offline RL (Hansen-Estruch et al., 2023; Park et al., 2025b) employing generative policies and extending them to online fine-tuning. To examine their behavior during online fine-tuning, we conducted an experiment with the challenging task `antmaze-giant-navigate` with FQL (Park et al., 2025b). The second plot of Figure 1 illustrates the maze, where the gray region marks the feasible paths, while the third plot shows the visitation frequency of the FQL agent during the first 100k interaction steps. It is seen that the agent stays mostly near the starting point and reaches the goal only via the upper path, ignoring other possible routes, yielding degraded performance as shown in the first plot in Figure 1. This behavior reflects an offline pre-trained policy that is overly confined to the dataset, which mainly contains the upper success route, resulting in limited exploration during online fine-tuning. In a strictly offline setting, such confinement to the data distribution is a primary design objective to ensure stability. But, considering the subsequent online learning, such confinement may not be the best strategy.

One could address this limitation by constructing a larger dataset with diverse data, but this incurs additional cost and time. Then, how can one induce diverse behavior from a given dataset without increasing the dataset size? To answer this question, we propose *perturbed cloning* during the offline pre-training phase, especially suited to flow-based policy learning. In our training scheme, we inject noise into flow matching, thereby driving the flow behavior model to extend its support beyond the coverage of the dataset to some reasonable extent. The policy is then distilled from this perturbed behavior model, which allows it to acquire behaviors spanning a broader action space. So learned policies can leverage this broader coverage during subsequent online fine-tuning, facilitating effective exploration to yield better performance, as shown in the rightmost plot of Figure 1. Details of the proposed method follow in the next section.

## 4 METHOD

In this section, we present Flow Matching with Injected Noise for Offline-to-Online RL (FINO), a novel method that utilizes flow policies within the offline-to-online RL framework. Our approach consists of two main components: (1) from the beginning of offline pre-training, we inject controlled noise into flow matching, encouraging the policy to explore a broader range of actions beyond those in the dataset; (2) in the online fine-tuning, we leverage this expanded action space for exploration, while introducing an entropy-guided sampling mechanism that dynamically balances exploration and exploitation according to the behavior of the policy.

### 4.1 NOISE INJECTION FOR FLOW MATCHING

Since our method builds directly on the flow matching formulation, we begin by presenting its conditional probability path. The rationale behind training flow matching as in Equation 3 lies in the design of its conditional probability path (Lipman et al., 2023):

$$p_t^{\text{FM}}(x|x_1) = \mathcal{N}(x|tx_1, (1 - (1 - \sigma_{\min})t)^2 I) \tag{6}$$

where $\sigma_{\min}$ is a sufficiently small constant ensuring that $p_1^{\text{FM}}(x|x_1)$ concentrates around the given data point.

In FQL (Park et al., 2025b), the variance is set to $\sigma_{\min} = 0$ in Equation 6, which reduces the training objective to Equation 4. With $\sigma_{\min} = 0$, the distribution collapses onto individual data points, leaving little coverage beyond the dataset itself. This narrow formulation shows clear limitations as shown in the previous section, as it restricts effective exploration during online fine-tuning. To overcome this limitation, we propose a noise-injected training scheme that retains the core objective of flow matching while enabling the model to learn a wider action space than point-wise matching:

$$\mathcal{L}_{\text{FINO}}(\theta) = \mathbb{E}_{\substack{s, a=x_1 \sim D, \\ x_0 \sim \mathcal{N}(0,I), \\ t \sim \text{Unif}([0,1])}} \left[ ||v_\theta(t, s, x_t + \epsilon_t) - (x_1 - (1 - \eta)x_0)||_2^2 \right], \quad \epsilon_t \sim \mathcal{N}(0, \alpha_t^2 I) \tag{7}$$

where $\alpha_t^2 = \left(\eta^2 - 2\eta\right)t^2 + (2\eta)t$ is the scheduled variance for some $\eta \in [0, 1]$, and $t$ is the interpolation time. Equation 7 reduces to the standard flow matching (Equation 4) when $\eta = 0$. The variance of the injected noise is non-negative for all $\eta \geq 0$ and $t \in [0, 1]$. Note that at $t = 0$, $\alpha_0^2 = 0$, and at $t = 1$, $\alpha_1^2 = \eta^2 > 0$.

**Proposition 1.** *For notational simplicity, we denote $(s_i, x_i^1)$ as $x_i$. Given a dataset $\mathcal{D} = \{x_i\}_{i=1}^N$, the proposed time-dependent noise injection $\epsilon_t \sim \mathcal{N}(0, \alpha_t^2 I)$ induces the following conditional probability paths of flow $\phi_t$:*

$$p_t^{FINO}(x|x_i) = \mathcal{N}\left(x \mid \mu_t(x_i) = tx_i, \Sigma_t(x_i) = (1 - (1 - \eta)t)^2 I\right),$$

*in which the mean $tx_i$ is equal to the mean induced from flow matching, and the variance $(1 - (1 - \eta)t)^2$ is greater than or equal to the variance induced from flow matching.*

**Theorem 1.** *Given a data $x_i$ from a dataset distribution and a noise $x_0$ from the base distribution, the conditional probability paths in Proposition 1 induce the unique conditional vector field that has the following form:*

$$v_t(x|x_i) = x_i - (1 - \eta)x_0.$$

*Then, for any dataset distribution, the marginal vector field $v_t(x)$ generates the marginal probability path $p_t(x)$, in other words, both $v_t(x)$ and $p_t(x)$ satisfy the continuity equation.*

Theorem 1 shows that FINO (Equation 7) yields a valid continuous normalizing flow, which means that the flow model trained by Equation 7 can generate samples close to those obtained by the behavior policy of the training dataset.

**Theorem 2.** *Suppose the cardinality of the dataset is finite, $\mathcal{D} = \{x_1, x_2, \ldots, x_N\}$ for some $N$, and data are independently and identically distributed (i.i.d.) sampled. The variance of the marginal probability path induced by FINO (Equation 7) is greater than or equal to that of the marginal probability path induced by the flow matching (FM) objective (Equation 3). For any time $t \in [0, 1]$,*

$$\text{Var}\left(X_t^{FINO}\right) \geq \text{Var}\left(X_t^{FM}\right), \quad X_t^{FINO} \sim p_t^{FINO}(x), \ X_t^{FM} \sim p_t^{FM}(x).$$

---

**Algorithm 1** FINO: Flow Matching with Injected Noise for Offline-to-Online RL

---

1: **Inputs:** flow matching policy $\beta_\theta$, one-step policy $\pi_\omega$, value function $Q_\phi$, candidate action samples $N_{\text{sample}}$, entropy update steps $N_\xi$
2: **while** in offline pre-training **do**
3:     Update $\omega, \theta$ based on Equation 5, 7 and update $\phi$ via TD loss
4: **end while**
5: **while** in online fine-tuning **do**
6:     Sample $N_{\text{sample}}$ candidate actions $\{a_1, a_2, \cdots, a_{N_{\text{sample}}}\} \sim \pi_\omega(s)$
7:     Compute sampling probability $p(i)$ using Equation 8
8:     Select $a$ from categorical distribution $p$
9:     Update $\omega, \theta$ based on Equation 5, 7 and update $\phi$ via TD loss
10:     **if** step mod $N_\xi$ == 0 **then**
11:         Estimate the entropy of policy $\mathcal{H}$
12:         Update $\xi$ using Equation 9
13:     **end if**
14: **end while**

---

Theorem 2 states that at time $t = 1$, the marginal probability path induced by FINO ($p_1^{\text{FINO}}(x)$) exhibits larger variance than flow matching ($p_1^{\text{FM}}(x)$). This means that the model trained by Equation 7 represents wider action regions than the flow matching model (Equation 3), making it more suitable for exploration. The proofs of Proposition 1 and Theorems 1, 2 are provided in Appendix C.

To illustrate the effect of our design, we conduct a simple toy experiment. We consider a setting with a fixed state and a two-dimensional action space, where the dataset is generated by sampling points inside four circular regions. Both flow matching and our proposed method are trained on the same dataset. As shown in Figure 2, flow matching predominantly focuses on the data points themselves, leading the trained actions to remain almost entirely within the dataset distribution. In contrast, our method with noise injection learns to cover a wider region of the action space. Notably, this expansion occurs in a reliable manner, remaining centered around the dataset and thereby providing a broader yet plausible coverage of the action space.

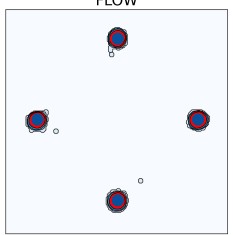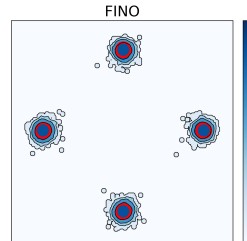

Figure 2: Toy example: blue contours represent the log-density of model samples; red circles denote the dataset.

The expanded flow model then guides the training of the one-step policy that interacts with the environment. As the one-step policy is trained under Equation 5, the expanded flow model enables action-value maximization over a broader region of the action space, which is then utilized for exploration during online fine-tuning. We provide a detailed explanation in Appendix D.

## 4.2 ENTROPY-GUIDED SAMPLING

After offline pre-training, the next step is to leverage the policy effectively during online fine-tuning, where the agent continues improving through direct interaction with the environment. With noise injection, the policy is trained to yield more diverse actions, each reflecting slightly different behaviors. To exploit this action diversity, the agent first samples multiple candidate actions for a given state using multiple base noises for the flow model. Now, we do not simply choose the action that maximizes action-value, which corresponds to exploitation. Instead, we construct a sampling distribution over the candidate actions based on their action-values as

$$p_{\text{sampling}}(i) = \frac{\exp(\xi \cdot Q_\phi(s, a_i))}{\sum_j \exp(\xi \cdot Q_\phi(s, a_j))}, \quad \forall i \in [1, \cdots, N_{\text{sample}}] \tag{8}$$

where $\xi$ is a temperature parameter. An actual action is drawn from $p_{\text{sampling}}$, so that even lower-value actions can be sampled for exploration purposes. A smaller $\xi$ produces a flatter distribution that promotes more uniform sampling and encourages exploration, whereas a larger $\xi$ sharpens the distribution and prioritizes greedy actions for exploitation.

Table 1: Performance of FINO and baselines across OGBench and D4RL tasks. Results show scores after offline pre-training and after online fine-tuning, averaged over 10 seeds with mean and 95% confidence intervals. D4RL antmaze and adroit aggregate six and four tasks, respectively, while OGBench reports results over five tasks (task names abbreviated by omitting the singletask suffix). Full results are presented in Table 4.

| Task | ReBRAC | Cal-QL | RLPD | IFQL | FQL | FINO |
|---|---|---|---|---|---|---|
| OGBench humanoidmaze-medium-navigate | $21_{\pm5} \rightarrow 3_{\pm3}$ | $0_{\pm0} \rightarrow 0_{\pm0}$ | $0_{\pm0} \rightarrow 1_{\pm1}$ | $59_{\pm7} \rightarrow 70_{\pm5}$ | $53_{\pm6} \rightarrow 61_{\pm2}$ | $50_{\pm7} \rightarrow \mathbf{97}_{\pm1}$ |
| OGBench humanoidmaze-large-navigate | $2_{\pm1} \rightarrow 1_{\pm0}$ | $0_{\pm0} \rightarrow 0_{\pm0}$ | $0_{\pm0} \rightarrow 0_{\pm0}$ | $11_{\pm3} \rightarrow 10_{\pm2}$ | $5_{\pm1} \rightarrow 10_{\pm3}$ | $6_{\pm2} \rightarrow \mathbf{33}_{\pm7}$ |
| OGBench antmaze-large-navigate | $85_{\pm3} \rightarrow \mathbf{99}_{\pm0}$ | $12_{\pm9} \rightarrow 12_{\pm8}$ | $0_{\pm0} \rightarrow 80_{\pm7}$ | $32_{\pm4} \rightarrow 72_{\pm6}$ | $81_{\pm2} \rightarrow 92_{\pm1}$ | $81_{\pm2} \rightarrow \mathbf{99}_{\pm0}$ |
| OGBench antmaze-giant-navigate | $35_{\pm7} \rightarrow \mathbf{96}_{\pm4}$ | $2_{\pm3} \rightarrow 0_{\pm0}$ | $0_{\pm0} \rightarrow 47_{\pm9}$ | $1_{\pm1} \rightarrow 0_{\pm0}$ | $16_{\pm6} \rightarrow 71_{\pm5}$ | $14_{\pm6} \rightarrow 79_{\pm0}$ |
| OGBench antsoccer-arena-navigate | $0_{\pm0} \rightarrow 0_{\pm0}$ | $0_{\pm0} \rightarrow 0_{\pm0}$ | $0_{\pm0} \rightarrow 2_{\pm1}$ | $33_{\pm4} \rightarrow 35_{\pm5}$ | $61_{\pm2} \rightarrow \mathbf{74}_{\pm4}$ | $57_{\pm3} \rightarrow \mathbf{77}_{\pm5}$ |
| OGBench cube-double-play | $9_{\pm2} \rightarrow 28_{\pm2}$ | $2_{\pm3} \rightarrow 0_{\pm0}$ | $0_{\pm0} \rightarrow 2_{\pm3}$ | $14_{\pm1} \rightarrow 40_{\pm2}$ | $31_{\pm3} \rightarrow 73_{\pm2}$ | $34_{\pm3} \rightarrow \mathbf{79}_{\pm2}$ |
| OGBench puzzle-4x4-play | $14_{\pm1} \rightarrow 29_{\pm5}$ | $2_{\pm3} \rightarrow 20_{\pm8}$ | $0_{\pm0} \rightarrow \mathbf{58}_{\pm11}$ | $26_{\pm2} \rightarrow 42_{\pm4}$ | $15_{\pm2} \rightarrow 45_{\pm8}$ | $19_{\pm3} \rightarrow \mathbf{56}_{\pm5}$ |
| D4RL antmaze | $80_{\pm5} \rightarrow 89_{\pm5}$ | $50_{\pm3} \rightarrow 89_{\pm2}$ | $0_{\pm0} \rightarrow 91_{\pm2}$ | $66_{\pm5} \rightarrow 79_{\pm5}$ | $80_{\pm4} \rightarrow \mathbf{95}_{\pm1}$ | $79_{\pm4} \rightarrow \mathbf{96}_{\pm1}$ |
| D4RL adroit | $21_{\pm2} \rightarrow 83_{\pm2}$ | $-0_{\pm0} \rightarrow -0_{\pm0}$ | $0_{\pm0} \rightarrow 73_{\pm5}$ | $18_{\pm1} \rightarrow 42_{\pm3}$ | $14_{\pm3} \rightarrow 100_{\pm6}$ | $13_{\pm2} \rightarrow \mathbf{112}_{\pm1}$ |

Since sample-efficient learning under a limited interaction budget is the primary objective of online fine-tuning, maintaining an appropriate balance between exploration and exploitation remains a critical challenge. However, relying on a fixed value of $\xi$ cannot adequately address the dynamics of the learning process. So, we adapt the sampling strategy to the behavior of the policy, using entropy of the policy ($\mathcal{H}$) as an indicator and adjusting $\xi$ accordingly:

$$\xi_{\text{new}} = \xi - \alpha_\xi[\mathcal{H} - \bar{\mathcal{H}}], \tag{9}$$

where $\bar{\mathcal{H}}$ is the target entropy, and $\alpha_\xi$ denotes the learning rate. By adapting its behavior according to the policy entropy, it properly controls the balance between exploration and exploitation throughout online fine-tuning. At inference time, the agent deterministically selects the action with the highest action-value, ensuring stable performance. The overall training pipeline is summarized in Algorithm 1.

### 4.3 PRACTICAL IMPLEMENTATION

We use FQL (Park et al., 2025b) as the backbone model, and accordingly the one-step policy, trained with Equation 5 is employed for environment interaction. Since this policy is obtained through distillation and action-value maximization, its distribution is intractable, making direct entropy computation infeasible. To address this, we follow prior work (Wang et al., 2024) and estimate entropy by sampling multiple actions from the same state and fitting them with a Gaussian Mixture Model (GMM). A detailed description of the computation procedure is provided in Appendix E.1.

Regarding hyperparameters, FINO involves two key parameters. For $\eta$, which determines the variance of the injected noise, we set its value based on the action range. Since all experimental environments use actions bounded within $[-1, 1]$, we fix $\eta = 0.1$. For $N_{\text{sample}}$, as the volume of the action space to explore grows significantly with the dimension, more samples are required to obtain a sufficiently diverse set of candidates for effective exploration. Therefore, we set the number of sampled actions to half of the action dimension. In implementation, we adopt a smooth shifted exponential schedule for $\alpha_t$ that satisfies the same boundary conditions, i.e., $\alpha_0^2 \approx 0$ and $\alpha_1^2 = \eta^2 > 0$, and we simply use the target vector $x_1 - x_0$, as we empirically observed no difference in performance. We note that these core hyperparameters remain fixed throughout the training process. Further implementation details and additional hyperparameter settings are provided in Appendix E and G.

## 5 EXPERIMENTS

In this section, we empirically demonstrate the effectiveness of FINO. To this end, we evaluate the proposed method across a range of challenging environments, comparing its performance against several baselines.

**Environments.** We primarily evaluate the performance of FINO on OGBench (Park et al., 2025a), a recently proposed benchmark that extends beyond the commonly used D4RL (Fu et al., 2020) by incorporating tasks with greater diversity and complexity. Although OGBench is originally introduced for benchmarking offline goal-conditioned RL, we adapt it to our setting by employing its

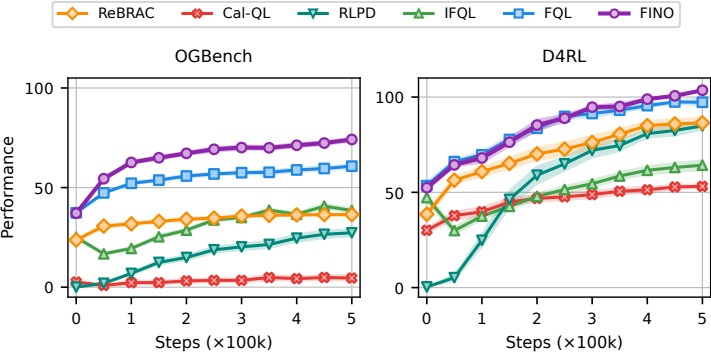

Figure 3: Aggregate performance across two benchmark domains. Each figure reports the averaged learning curves over the common environments within the respective domain. Full results are presented in Figures 9 and 10.

single-task variant, where each goal is treated as an independent task. We also include results on the widely adopted D4RL benchmark, which remains a common benchmark in offline-to-online RL.

**Baselines.** We consider the following baselines for comparison: (1) ReBRAC (Tarasov et al., 2023) is a Gaussian policy-based method, has demonstrated strong performance across offline RL and the offline-to-online RL setting. (2) Cal-QL (Nakamoto et al., 2023) is a representative offline-to-online RL algorithm that extends CQL (Kumar et al., 2020) to the offline-to-online setting. (3) RLPD (Ball et al., 2023) is an online RL algorithm initialized with an offline dataset, which achieves superior performance despite relying solely on online training. Since prior work has provided limited investigation of flow matching and denoising diffusion in the offline-to-online RL setting, we additionally construct flow matching variants of existing algorithms to provide a meaningful point of comparison. (4) IFQL, introduced in the FQL paper (Park et al., 2025b), is an adaptation of IDQL (Hansen-Estruch et al., 2023) to the flow matching setting. Similar to our approach, it samples multiple actions from a single state and selects one for execution. (5) FQL (Park et al., 2025b), described in Section 2, serves as the backbone algorithm upon which our method is built.

**Evaluation.** For all baselines, we report results using the same experimental protocol, consisting of 1M offline pre-training steps followed by 500K online fine-tuning steps. To assess the performance gain during online fine-tuning, we present both the results immediately after offline pre-training and those obtained at the end of online fine-tuning. All experiments are averaged over 10 random seeds, and we report the mean and 95% confidence intervals. The best-performing results are highlighted in bold when they fall within 95% of the best performance.

**Results.** Table 1 summarizes the results across a total of 45 tasks, aggregated by task category. Overall, FINO consistently achieves strong performance across a diverse range of environments. Crucially, this is achieved without degrading offline performance, as our method learns the model that preserves the mean of the probability path while increasing variance from the offline dataset, supported by Theorem 2. When compared to ReBRAC (Tarasov et al., 2023), we observe that although ReBRAC exhibits strong performance on environments such as `antmaze`, it struggles to effectively learn in more complex and challenging `humanoidmaze` environments due to the inherent limitations of its conventional policy. The comparison with IFQL underscores that action candidate sampling alone is insufficient to explain the performance improvement. In addition, when compared to the backbone algorithm FQL (Park et al., 2025b), the results highlight the effectiveness of our method during online fine-tuning, where FINO demonstrates both efficient exploration and a balanced trade-off between exploration and exploitation. The improvements observed in the `navigate` environments further suggest that FINO is well suited to environments where effective exploration is critical.

In addition to the tabular summary, we provide aggregate learning curves by benchmark in Figure 3 to visualize the progression of performance over training steps. Consistent with the results in Table 1, the figure shows that FINO consistently outperforms the baselines throughout training. In particular, the experiments on OGBench demonstrate that, despite starting from the same performance as the backbone algorithm, our method achieves stronger improvements, underscoring its high sample efficiency.

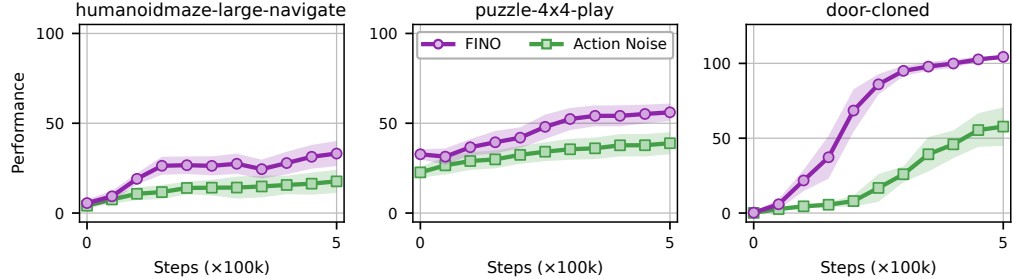

Figure 4: Comparison between FINO and the direct action noise injection baseline. Each plot shows results aggregated over five tasks and averaged across 10 seeds, with shaded regions indicating 95% confidence intervals.

## 6 DISCUSSION

### 6.1 IMPACT OF NOISE INJECTION POINT

One of the key components of the proposed method is the injection of noise into the flow matching objective, which expands the action space and enables more efficient exploration during online fine-tuning. To evaluate this design choice, we compare our approach with a simpler alternative that promotes exploration by injecting Gaussian noise into the actions generated by the policy rather than into the flow matching objective (denoted as *Action Noise*). For a fair comparison, we retain other components such as the action-candidate mechanism and entropy guidance in this baseline as well.

The results in Figure 4 demonstrate that the noise injection strategy of the proposed method yields a notable performance gain. Consistent improvement is observed across both navigation and manipulation environments, indicating that the proposed noise injection scheme remains effective across task categories. This difference arises because, as mentioned in Section 4.1, the proposed method enables the one-step policy to maximize action-value over a broader action space, rather than simply adding noise to the action. Notably, the results on `door-cloned` show that, when compared with the backbone algorithm FQL (whose performance is approximately 100), simply adding noise to the action fails to facilitate exploration and can even degrade performance. These findings highlight that the proposed noise injection method serves as an effective approach for promoting exploration during online fine-tuning. Additional analyses of various noise injection strategies are provided in Appendix D.

### 6.2 COMPARISON WITH ENTROPY-REGULATED NOISE SCALING

In Section 4.2, we introduce an entropy-based guidance method for action sampling. This approach enables the policy to achieve a balanced trade-off between exploration and exploitation during online fine-tuning, which is critical under a limited online budget. Since previous studies (Haarnoja et al., 2018; Wang et al., 2024) have also employed entropy to regulate this balance, we compare our method with an alternative entropy-driven strategy (denoted as *ER-Noise*) to evaluate its effectiveness. Specifically, instead of using entropy to guide the action sampling, the baseline replaces it with a simpler approach that scales the Gaussian noise based on the entropy. The noise is then directly added to the action, allowing the action to be adjusted according to the entropy of the policy.

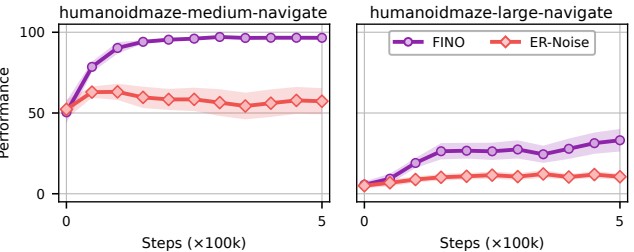

Figure 5: Comparison between FINO and the entropy-regulated noise scaling baseline. Full results are presented in Table 5.

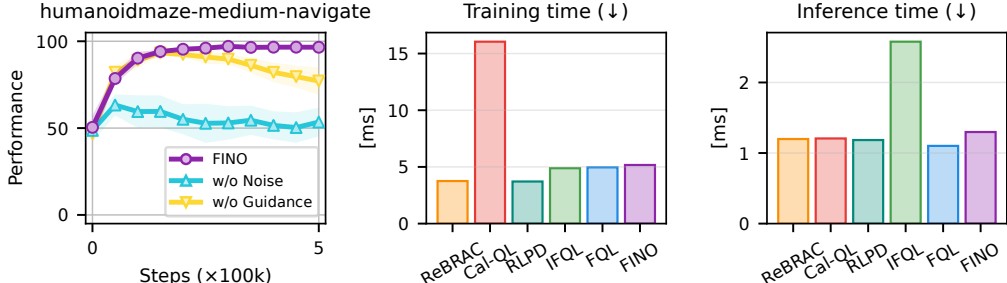

Figure 6: Comparison of performance and computational efficiency. The left figure shows the learning curve on the `humanoidmaze-medium-navigate` task, averaged over five tasks with 10 random seeds, with shaded regions denoting 95% confidence intervals. The middle and right figures report the training and inference time per step of each baseline. Full results are presented in Table 5.

Figure 5 shows that the proposed method significantly outperforms entropy-based noise scaling by effectively balancing exploration and exploitation through sampling aligned with the entropy of the policy. In particular, despite the inherent difficulty of finding relevant actions in high-dimensional action spaces such as `humanoidmaze`, the proposed method successfully identifies appropriate actions in such settings. However, the performance of *ER-Noise* indicates that mere noise scaling guided by entropy is insufficient for selecting actions consistent with the policy. These findings thus confirm that the proposed method effectively leverages entropy to enable sample-efficient learning during online fine-tuning.

### 6.3 ANALYSIS OF NOISE INJECTION AND ENTROPY-GUIDED SAMPLING

In our proposed method, we incorporate two key components, namely noise injection during offline pre-training and entropy-guided sampling during online fine-tuning. To clarify the contribution of each element, we design controlled experiments that reflect its intended role. In particular, we first examine the case without injected noise (`w/o Noise`), where the procedure reduces to the same formulation as Equation 4 while still retaining entropy-guided sampling. We then consider the case without entropy guidance (`w/o Guidance`), in which the sampling process still produces action candidates, but the selection is restricted to the one with the highest action-value, thereby excluding the entropy-based balancing mechanism.

The left plot of Figure 6 demonstrates that both components are indispensable to the effectiveness of our method. Noise injection, in particular, proves to be especially critical. This is because, without it, the offline pre-training is limited to actions contained in the offline dataset. Even when action candidates are generated, this restriction makes them lack diversity, which in turn leads to insufficient exploration and thereby reduces overall performance. In the absence of entropy guidance, the performance deteriorates in later stages, as the training process fails to maintain an appropriate balance between exploration and exploitation. These observations suggest that both noise injection and entropy-guided sampling play an important role in enabling sample-efficient learning in the offline-to-online RL setting.

### 6.4 TRAINING AND INFERENCE EFFICIENCY

Since computational cost is also an important factor in methods employing generative models, we compare our algorithm with the baselines on `humanoidmaze-medium`, where it achieves the largest performance improvement. The middle and right plots of Figure 6 present the training time and inference time, respectively. The results show that although additional components such as entropy estimation and action candidate sampling slightly increase training time relative to the backbone algorithm, this increase is negligible when compared to algorithms such as Cal-QL, leaving overall training efficiency largely unaffected. Regarding inference time, our method requires fewer samples than baselines such as IFQL, demonstrating that the additional computation does not impose a significant overhead.

## 6.5 Effect of Action Sample Size ($N_{\text{sample}}$)

The proposed method injects noise into the flow matching objective and samples action from a set of action candidates to utilize the expanded policy. Since the size of the exploration space increases with the action dimension of the task, we set the hyperparameter $N_{\text{sample}}$, which determines the number of action candidates, to half of the action dimension. To examine the impact of this hyperparameter, we evaluate performance across 6 environments with action dimensions greater than 10 while varying $N_{\text{sample}}$.

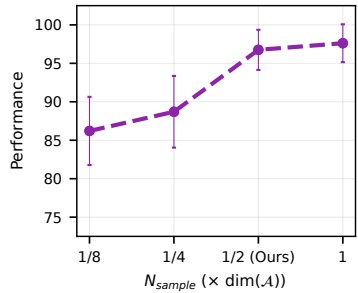

Figure 7: Comparison of performance with varying $N_{\text{sample}}$.

As shown in Figure 7, performance generally improves as the number of action candidates increases, but the marginal gains diminish beyond a certain point. Since $N_{\text{sample}}$ directly affects inference cost, we adopt the choice of setting it to half of the action dimension, achieving a practical balance between performance and computational overhead.

## 7 Related Work

**Offline-to-Online Reinforcement Learning.** Offline-to-online RL is a framework that learns through two stages: offline pre-training and online fine-tuning (Lee et al., 2022; Zhang et al., 2023; Wang et al., 2023a; Nakamoto et al., 2023; Zhou et al., 2024). The goal is to improve an agent, initially trained on offline data, by allowing it to further refine through environment interaction. The simplest way to train within this framework is to employ the same offline RL algorithm for both offline pre-training and online fine-tuning (Lyu et al., 2022; Wu et al., 2022; Tarasov et al., 2023). However, this strategy inherits the conservative nature of offline RL methods, which restricts exploration during online fine-tuning (Yu & Zhang, 2023; Luo et al., 2024; Zhang et al., 2024; Kim et al., 2025). Several prior studies have attempted to ease this conservatism and support more effective online fine-tuning (Wang et al., 2023a; Nakamoto et al., 2023). Still, because these approaches remain conservative, they fail to provide sample efficiency through effective exploration (Shin et al., 2025). In contrast, our method leverages the expressivity of generative models to encourage exploration and regulates it with entropy-guided sampling, thereby achieving high sample efficiency.

**Reinforcement Learning with Generative Models.** Recent advances in generative models such as denoising diffusion (Sohl-Dickstein et al., 2015; Ho et al., 2020) and flow matching (Lipman et al., 2023; Albergo & Vanden-Eijnden, 2023) have spurred growing interest in applying these techniques to RL. Among these efforts, research that employs generative models as policies has shown strong results in both offline (Wang et al., 2023b; Hansen-Estruch et al., 2023; Kang et al., 2023; Zhang et al., 2025; Kim et al., 2024a;b; Fang et al., 2025; Park et al., 2025b; Chae et al., 2026) and online (Wang et al., 2024; Psenka et al., 2024; Ding et al., 2024) settings by framing the policy as a state-conditioned generative model. Within the offline-to-online RL framework, there have been studies that exploit the expressive capacity of diffusion models for data augmentation (Liu et al., 2024; Huang et al., 2025). However, no prior work has leveraged such expressivity directly as a policy in this setting. In contrast, our work harnesses the generative model for exploration, demonstrating a method that achieves strong sample efficiency in offline-to-online RL.

## 8 Conclusion

In this work, we propose FINO, a novel approach that leverages the expressivity of flow matching through noise injection and enhances online fine-tuning via entropy-guided sampling. Noise injection, applied to the offline pre-training, broadens the action space and yields a stronger initialization for exploration, while entropy-guided sampling adapts to the policy's evolving behavior to maintain a workable exploration–exploitation balance. FINO achieves sample-efficient learning across diverse and challenging benchmarks while maintaining modest computational cost. Beyond empirical gains, our study highlights how flow matching can be effectively utilized to address the challenges of offline-to-online RL, and we believe this line of work opens new directions for harnessing generative policies in advancing the broader offline-to-online RL paradigm.

ACKNOWLEDGEMENTS

This work was supported in part by the Institute of Information & Communications Technology Planning & Evaluation (IITP) grant funded by the Korea government (MSIT) (No. RS-2022-II220469, Development of Core Technologies for Task-oriented Reinforcement Learning for Commercialization of Autonomous Drones, 50%) and in part by the National Research Foundation of Korea (NRF) grant funded by the Korea government (MSIT) (No. RS-2025-00557589, Generative Model Based Efficient Reinforcement Learning Algorithms for Multi-modal Expansion in Generalized Environments, 50%). We would like to thank Woohyeon Byeon for providing valuable insights into Theorem 2.

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

## A  LIMITATIONS

The entropy of the policy is approximated using a Gaussian Mixture Model (GMM) (Huber et al., 2008), which incurs computational overhead and remains an approximation rather than an exact calculation (Wang et al., 2024). Future work could focus on developing entropy estimation methods that are both computationally efficient and more precise, or on exploring alternative metrics that capture policy behavior beyond entropy. In addition, while our method directly addresses the challenges of exploration and the exploration–exploitation trade-off in offline-to-online RL, the issue of distribution shift remains. Addressing this challenge constitutes another promising direction for future research.

## B  THE USE OF LARGE LANGUAGE MODELS (LLMs)

Large Language Models are used solely to aid and polish the writing. They are not involved in research ideation, methodological design, or experimental analysis.

## C  THEORETICAL PROOFS

**Proposition 1.** *For notational simplicity, we denote $(s_i, x_i^1)$ as $x_i$. Given a dataset $\mathcal{D} = \{x_i\}_{i=1}^N$, the proposed time-dependent noise injection $\epsilon_t \sim \mathcal{N}(0, \alpha_t^2 I)$ induces the following conditional probability paths of flow $\phi_t$:*

$$p_t^{FINO}(x|x_i) = \mathcal{N}\left(x \mid \mu_t(x_i) = tx_i, \, \Sigma_t(x_i) = (1 - (1 - \eta)t)^2 I\right),$$

*in which the mean $tx_i$ is equal to the mean induced from flow matching, and the variance $(1 - (1 - \eta)t)^2$ is greater than or equal to the variance induced from flow matching.*

*Proof.* We consider time-dependent conditional probability paths of flow $\phi_t(x)$ as follows:

$$p_t(x|x_i) = \mathcal{N}\left(x|\mu_t(x_i), \sigma_t^2(x_i)I\right) \tag{10}$$

Following the flow matching (Lipman et al., 2023), we set the time-dependent mean and variance as

$$\mu_t(x_i) = tx_i, \quad \sigma_t(x_i) = 1 - (1 - \sigma_{\min})t,$$

where $\sigma_{\min}$ is a negligible small constant. The conditional probability paths provide the following canonical transformation for Gaussian distribution, i.e., the flow conditioned on $x_i$:

$$\begin{aligned}
\phi_t(x) &= \sigma_t(x_i)x + \mu_t(x_i) = (1 - (1 - \sigma_{\min})t)x + tx_i \\
&\approx (1 - t)x + tx_i
\end{aligned}$$

By injecting the introduced time-dependent noise $\epsilon_t$, we obtain the perturbed flow

$$\tilde{\phi}_t(x) = tx_i + (1 - t)x + \epsilon_t,$$

where $x$ is distributed as normal distribution $\mathcal{N}(0, I)$.

Since the injected noise is designed as a Gaussian distribution $\mathcal{N}(0, \alpha_t^2 I)$ and is independent over distribution 10, the perturbed flow can be expressed as the sum of two Gaussian distributions. This leads to the following conditional probability paths:

$$\begin{aligned}
\tilde{p}_t(x|x_i) &= \mathcal{N}\left(x|tx_i, ((1 - t)^2 + \alpha_t^2)I\right) \\
&= \mathcal{N}\left(x|tx_i, ((1 - t)^2 + (\eta^2 - 2\eta)t^2 + 2\eta t)I\right) \\
&= \mathcal{N}\left(x|tx_i, (1 - (1 - \eta)t)^2 I\right)
\end{aligned}$$

as desired. $\square$

**Theorem 1.** *Given a data $x_i$ from a dataset distribution and a noise $x_0$ from the base distribution, the conditional probability paths in Proposition 1 induce the unique conditional vector field that has the following form:*

$$v_t(x|x_i) = x_i - (1 - \eta)x_0.$$

*Then, for any dataset distribution, the marginal vector field $v_t(x)$ generates the marginal probability path $p_t(x)$, in other words, both $v_t(x)$ and $p_t(x)$ satisfy the continuity equation.*

*Proof.* The conditional probability path $p_t(x_t|x_i) = p_t(\phi_t(x_0)|x_i)$ provides the canonical transformation of Gaussian distribution as the perturbed flow $\phi_t(x)$ conditioned on $x_i$:

$$x_t = \phi_t(x_0) = tx_i + (1 - (1 - \eta)t)x_0,$$

its derivative is the vector field that generates the flow $\phi_t$ by the definition of vector fields of continuous normalizing flow (Lipman et al., 2023).

$$\frac{d}{dt}\phi_t(x_0) = \frac{d}{dt}(tx_i + (1 - (1 - \eta)t)x_0) = x_i - (1 - \eta)x_0,$$

which is the same result as one from Theorem 3 of (Lipman et al., 2023):

$$
\begin{aligned}
v_t(x_t|x_i) &= \frac{\sigma_t'(x_i)}{\sigma_t(x_i)}(x_t - \mu_t(x_i)) + \mu_t'(x_i), \\
&= \frac{-(1-\eta)}{1-(1-\eta)t}(x_t - tx_i) + x_i \\
&= \frac{-(1-\eta)x_t + (1-\eta)tx_i + (1-(1-\eta)t)x_i}{1-(1-\eta)t} \\
&= \frac{-(1-\eta)(tx_i + (1-(1-\eta)t)x_0) + x_i}{1-(1-\eta)t} \\
&= \frac{(1-(1-\eta)t)x_i - (1-\eta)(1-(1-\eta)t)x_0}{1-(1-\eta)t} \\
&= x_i - (1-\eta)x_0
\end{aligned}
$$

where $f'$ denotes the derivative w.r.t interpolation time $t$, and $\sigma_t(x_i)I = \Sigma_t(x_i)$ and $\mu_t(x_i)$ are the covariance and the mean of $p_t(x_t|x_i)$, respectively. This means that the Gaussian probability path $p_t(x_t|x_i)$ induces the unique conditional vector field $v_t(x_t|x_i) = x_i - (1-\eta)x_0$.

Since the conditional probability path $p_t(x_t|x_i)$ has the form of Gaussian probability distribution, and the unique vector field $v_t(x_t|x_i)$ generates the perturbed flow $\phi_t(x)$ conditioned on $x_i$, we can use the definition of the marginal probability paths $p_t(x_t)$ and the marginal vector field $v_t(x_t)$ in flow matching (Lipman et al., 2023):

$$
p_t(x) = \int p_t(x_t|x_i)q(x_i)dx_i, \quad p_1(x_i) \approx q(x_i)
$$

$$
v_t(x) = \int v_t(x|x_i)\frac{p_t(x|x_i)q(x_i)}{p_t(x_i)}dx_i,
$$

where $q$ is a dataset distribution, both marginal probability paths $p_t(x_t)$ and vector field $v_t(x_t)$ satisfy the continuity equation (Villani et al., 2008) (refer to Theorem 1 of Lipman et al. (2023)). $\qquad\square$

**Theorem 2.** *Suppose the cardinality of the dataset is finite, $\mathcal{D} = \{x_1, x_2, \ldots, x_N\}$ for some $N$, and data are independently and identically distributed (i.i.d.) sampled. The variance of the marginal probability path induced by FINO (Equation 7) is greater than or equal to that of the marginal probability path induced by the flow matching (FM) objective (Equation 3). For any time $t \in [0, 1]$,*

$$
\mathrm{Var}\left(X_t^{FINO}\right) \geq \mathrm{Var}\left(X_t^{FM}\right), \quad X_t^{FINO} \sim p_t^{FINO}(x), \ X_t^{FM} \sim p_t^{FM}(x).
$$

*Proof.* For notational simplicity, given a data $x_i$, let $p_t^{\text{FINO}}(x)$ and $p_t^{\text{FINO}}(x|x_i)$ be the marginal and conditional probability paths induced by equation 7, and $p_t^{\text{FM}}(x)$ and $p_t^{\text{FM}}(x|x_i)$ be them induced by equation 3.

Since data are i.i.d. sampled, we can write the marginal probability paths as follows:

$$
p_t^{\text{FINO}}(x) = \int p_t^{\text{FINO}}(x|x_i)q(x_i)dx_i = \frac{1}{N}\sum_{i=1}^{N} p_t^{\text{FINO}}(x|x_i)
$$

$$
p_t^{\text{FM}}(x) = \int p_t^{\text{FM}}(x|x_i)q(x_i)dx_i = \frac{1}{N}\sum_{i=1}^{N} p_t^{\text{FM}}(x|x_i)
$$

To simplify notation, we denote random variables as $X_t$ and $X_t^i$ that follow a marginal distribution $p_t(x)$ and $p_t(x|x_i)$, respectively, given a data $x_i$. Assume the distributions $p_t(x)$ and $p_t(x|x_i)$ have identity covariances, then, for fixed $t$, the random variable $X_t$ from the marginal probability path $p_t(x)$ has the variance as follows (the following equation can apply to both $p_t^{\text{FINO}}(x)$ and $p_t^{\text{FM}}(x)$

since their conditional probability paths are isotropic Gaussian distributions):

$$
\begin{aligned}
\mathrm{Var}\,(X_t) &= \mathbb{E}_{p_t}[X_t^2] - \mathbb{E}_{p_t}[X_t]^2 = \int p_t(x)x^2 dx - \left(\int p_t(x)x dx\right)^2 \\
&= \int \left(\frac{1}{N}\sum_{i=1}^{N} p_t^i(x)\right)x^2 dx - \left(\int \left(\frac{1}{N}\sum_{i=1}^{N} p_t^i(x)\right)x dx\right)^2 \\
&= \frac{1}{N}\sum_i \int p_t^i(x)x^2 dx - \left(\frac{1}{N}\sum_i \int p_t^i(x)x dx\right)^2 \\
&= \frac{1}{N}\sum_i \mathbb{E}_{p_t^i}[X_t^2] - \left(\frac{1}{N}\sum_i \mathbb{E}_{p_t^i}[X_t]\right)^2 \\
&= \frac{1}{N^2}\left(\sum_i N\mathbb{E}_{p_t^i}[X_t^2] - \sum_{i=1}^{N}\sum_{j=1}^{N}\mathbb{E}_{p_t^i}[X_t]\mathbb{E}_{p_t^j}[X_t]\right) \\
&= \frac{1}{N^2}\left(\sum_i N\mathbb{E}_{p_t^i}[X_t^2] - \sum_i \mathbb{E}_{p_t^i}[X_t]^2 - \sum_i \sum_{j:j\neq i}\mathbb{E}_{p_t^i}[X_t]\mathbb{E}_{p_t^j}[X_t]\right) \\
&= \frac{1}{N^2}\left(\sum_i N\mathbb{E}_{p_t^i}[X_t^2] - \sum_i N\mathbb{E}_{p_t^i}[X_t]^2 + \sum_i (N-1)\mathbb{E}_{p_t^i}[X_t]^2 - \sum_i \sum_{j:j\neq i}\mathbb{E}_{p_t^i}[X_t]\mathbb{E}_{p_t^j}[X_t]\right) \\
&= \frac{1}{N^2}\left(\sum_i N\mathrm{Var}(X_t^i) + \sum_i (N-1)\mathbb{E}_{p_t^i}[X_t]^2 - \sum_i \sum_{j:j\neq i}\mathbb{E}_{p_t^i}[X_t]\mathbb{E}_{p_t^j}[X_t]\right)
\end{aligned}
$$

Using the equation above, the variance of the marginal probability path $p_t^{\mathrm{FINO}}$ can be rewritten as

$$
\mathrm{Var}(X_t^{\mathrm{FINO}}) = \frac{1}{N^2}\left(\sum_{i=1}^{N} N\sigma_t^{i,\mathrm{FINO}}d + \sum_i (N-1)\mu_t^{i,\mathrm{FINO}} - \sum_i \sum_{j:j\neq i}\mu_t^{i,\mathrm{FINO}}\mu_t^{j,\mathrm{FINO}}\right), \quad (11)
$$

where $d$ is the dimension of data $x_i$, given data $x_i$, $\sigma_t^{i,\mathrm{FINO}}$ is the variance of the conditional probability path $p_t^{\mathrm{FINO}}(x|x_i)$, and $\mu_t^{i,\mathrm{FINO}}$ is the mean of the path.

By the same argument, the variance of the marginal probability path of FM $p_t^{\mathrm{FM}}$

$$
\mathrm{Var}(X_t^{\mathrm{FM}}) = \frac{1}{N^2}\left(\sum_{i=1}^{N} N\sigma_t^{i,\mathrm{FM}}d + \sum_i (N-1)\mu_t^{i,\mathrm{FM}} - \sum_i \sum_{j:j\neq i}\mu_t^{i,\mathrm{FM}}\mu_t^{j,\mathrm{FM}}\right), \quad (12)
$$

where $d$ is the dimension of data $x_i$, given data $x_i$, $\sigma_t^{i,\mathrm{FM}}$ is the variance of the conditional probability path $p_t^{\mathrm{FM}}(x|x_i)$, and $\mu_t^{i,\mathrm{FM}}$ is the mean of the path.

From Proposition 1, we already have $\mu_t^{i,\mathrm{FINO}} = \mu_t^{i,\mathrm{FM}}$ and $\sigma_t^{i,\mathrm{FINO}} \geq \sigma_t^{i,\mathrm{FM}}$, by subtracting equation 11 from equation 12, then we obtain

$$
\mathrm{Var}(X_t^{\mathrm{FINO}}) - \mathrm{Var}(X_t^{\mathrm{FM}}) = \frac{1}{N^2}\sum_{i=1}^{N} Nd\left(\sigma_t^{i,\mathrm{FINO}} - \sigma_t^{i,\mathrm{FM}}\right) \geq 0
$$

$\square$

# D   ANALYSIS OF NOISE INJECTION

In Section 4.1, we describe a training approach that injects noise into the flow matching objective, enabling the model to learn over a broader action space. In this section, we discuss alternative noise injection strategies that can be applied to the flow matching objective and illustrate their effects through a toy example. Throughout this section, we assume the use of zero-mean noise $\epsilon \sim p_{\text{noise}}$ (e.g., Gaussian Noise).

## D.1   CASE 1: INJECTING NOISE TO VELOCITY TARGET

Adding noise to the target velocity in Equation 4 is the simplest form of noise injection. The flow matching objective with the added noise can be written as follows:

$$\mathcal{L}_\pi(\theta) = \mathbb{E}_{\substack{x_0 \sim \mathcal{N}(0,I), \\ s, a = x_1 \sim D, \\ t \sim \text{Unif}([0,1]), \\ \epsilon \sim p_{\text{noise}}}} \left[ ||v_\theta(t, s, x_t) - (x_1 - x_0) - \epsilon||_2^2 \right]$$

$$= \mathbb{E} \left[ ||v_\theta(t, s, x_t) - (x_1 - x_0)||_2^2 - 2 \left( v_\theta(t, s, x_t) - (x_1 - x_0) \right)^\top \epsilon + ||\epsilon||_2^2 \right].$$

Since the noise has zero mean ($\mathbb{E}_{\epsilon \sim p_{\text{sample}}}[\epsilon] = 0$), the second term becomes zero in expectation. Furthermore, because $\epsilon$ is independent of $\theta$, the last term $||\epsilon||_2^2$ is a constant with respect to $\theta$, so it does not contribute to the gradient during optimization. As a result, the total gradient of the objective is identical to that of the original flow matching objective in Equation 4, which means that training proceeds in exactly the same way in expectation.

### D.2 CASE 2: INJECTING NOISE TO POLICY ACTION

A straightforward way to encourage exploration is to inject noise directly to the actions generated by the policy. However, as shown in Section 6.1, this approach leads to limited improvement during online fine-tuning. This difference stems from the training process of the one-step policy, as described in Section 4.1, which is the component that directly interacts with the environment.

The one-step policy is trained with the following objective:

$$\mathcal{L}_\pi(\omega) = \mathbb{E}_{\substack{s \sim D, \\ z \sim \mathcal{N}(0,I), \\ a_\omega(s,z) \sim \pi_\omega}} \big[ -Q_\phi(s, a_\omega(s,z)) + \alpha \|a_\omega(s,z) - a_\theta(s,z)\|_2^2 \big].$$

It distills the flow model trained from the offline dataset while simultaneously maximizing action-value through the value function. Since our algorithm employs the flow model trained with the expanded action space from Equation 7, the resulting one-step policy learns to explore regions that are more informative for improving action-values. In contrast, simply adding noise to the action ignores action-value information, making it an inherently less efficient exploration strategy.

To illustrate this difference, we conduct a comparative experiment in a toy example. In this setting, the state is fixed, and the action is two-dimensional, corresponding to the x- and y- axes in the figure. The dataset is sampled from a Gaussian distribution centered at the origin, and the reward increases monotonically toward the right. We train two flow models using Equations 4 and 7, respectively, and each flow model is then used to train a separate one-step policy via Equation 5. The baseline that uses the flow model trained with Equation 4 and injects Gaussian noise directly into the action output is referred to as the *Action Noise*. The approach based on the flow model trained with Equation 7 corresponds to our proposed method, which does not apply any modification to the action output.

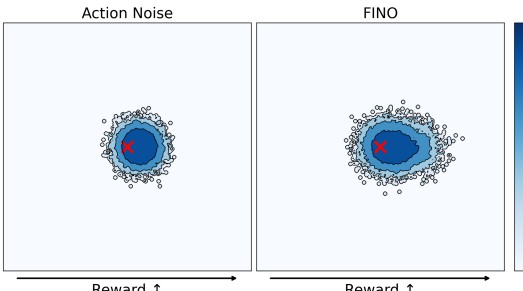

Figure 8: Action samples of two one-step policies: blue contours indicate the log-density of the sampled actions, and red × marks denote the centers of the dataset.

Figure 8 shows that, since the reward increases toward the right, the action samples generated by both policies shift rightward relative to the dataset. While adding noise directly to the actions causes the samples to spread in random directions, FINO guides the samples toward regions with higher action-values. This is because the method leverages both the expanded action space and the action-value maximization objective. As a result, the one-step policy is guided toward a more informed learning direction, leading to more effective exploration during online fine-tuning and ultimately explaining the superior performance of our approach.

# E  IMPLEMENTATION DETAILS

We implement our proposed method, FINO, based on the official implementation of FQL[1]. In FINO, the entropy estimation module is adapted from the official DACER implementation[2]. For all baselines except Cal-QL, we adopt the components provided in the FQL implementation, while for Cal-QL[3] we rely on its official implementation.

## E.1  ENTROPY ESTIMATION

In our proposed algorithm, entropy-guided sampling requires an estimate of policy entropy. However, since the one-step policy training objective combines behavior cloning from flow policy with action-value maximization, the entropy cannot be derived in closed form. To address this, we adopt an estimation strategy introduced in prior work (Wang et al., 2024).

To compute the policy entropy, we employ a Gaussian Mixture Model (GMM) as an approximation of the action distribution. A GMM represents complex data distributions by combining multiple Gaussian components. Formally, the likelihood of a sample under a GMM is defined as a mixture of Gaussian densities:

$$p(a) = \sum_{k=1}^{K} \pi_k \mathcal{N}(a|\mu_k, \Sigma_k) \tag{13}$$

where $K$ denotes the number of Gaussian components and $\pi_k \in [0, 1]$ is the mixing coefficient that specifies the probability of selecting the $k$-th Gaussian, satisfying $\sum_{k=1}^{K} \pi_k = 1$.

Training a GMM corresponds to estimating its parameters such that the likelihood of the given data is maximized. To approximate the action distribution of the policy using a GMM, we first sample multiple actions $(a^1, a^2, \cdots, a^N)$ from the policy for each state. We then fit the GMM to these samples using the Expectation-Maximization (EM) algorithm. The EM algorithm alternates between two iterative phases, namely the expectation step and the maximization step. In the expectation step, the latent probabilities required to compute the likelihood are estimated:

$$\gamma(z_k^n) = \frac{\pi_k \mathcal{N}(a^n|\mu_k, \Sigma_k)}{\sum_{i=1}^{K} \pi_i \mathcal{N}(a^n|\mu_i, \Sigma_i)} \tag{14}$$

where $\gamma(z_k^n)$ denotes that under the current parameter estimates, the observed data $a^n$ come from the $k$-th component of the probability. In the maximization step, the GMM parameters are updated based on these probabilities:

$$\pi_k = \frac{1}{N} \sum_{n=1}^{N} \gamma(z_k^n), \quad \mu_k = \frac{\sum_{n=1}^{N} \gamma(z_k^n) \cdot a^n}{\sum_{n=1}^{N} \gamma(z_k^n)}, \quad \Sigma_k = \frac{\sum_{n=1}^{N} \gamma(z_k^n)(a^n - \mu_k)(a^n - \mu_k)^{\mathrm{T}}}{\sum_{n=1}^{N} \gamma(z_k^n)} \tag{15}$$

By repeating these two steps until convergence, we obtain a GMM that approximates the action distribution of the policy.

The entropy of the fitted GMM is then computed following the approach established in prior work (Huber et al., 2008):

$$\mathcal{H} \approx \sum_{k=1}^{K} \pi_k \cdot \left( -\log \pi_k + \frac{1}{2} \log((2\pi e)^d |\Sigma_k|) \right) \tag{16}$$

where $d$ denotes the dimensionality of the action space. The entropy estimate of the policy is obtained by averaging this quantity across a batch of sampled states. In practice, we set the number of mixture components to $K = 3$, which we found sufficient across all tasks. The number of states used for entropy estimation is determined by the batch size, and the number of actions sampled per state is fixed at 200, following prior work (Wang et al., 2024).

---

[1]https://github.com/seohongpark/fql
[2]https://github.com/happy-yan/DACER-Diffusion-with-Online-RL
[3]https://github.com/nakamotoo/Cal-QL

# F    EXPERIMENTAL DETAILS

## F.1    BENCHMARKS

We conduct experiments on 35 tasks from OGBench (Park et al., 2025a) and 10 tasks from D4RL (Fu et al., 2020). For OGBench, we adopt single-task variants (`singletask`) provided in the benchmark and configure them to fit the offline-to-online RL framework. Each environment consists of five distinct tasks, each defined by a different goal. In our experiments, we use the following 7 environments and datasets:

- humanoidmaze-medium-navigate-v0
- humanoidmaze-large-navigate-v0
- antmaze-large-navigate-v0
- antmaze-giant-navigate-v0
- antsoccer-arena-navigate-v0
- cube-double-play-v0
- puzzle-4x4-play-v0

In `humanoidmaze`, the objective is to control a humanoid robot with a 21-dimensional action space to reach the designated goal. In `antmaze` and `antsoccer`, the agent controls a quadrupedal robot with an 8-dimensional action space to navigate the goal; in `antsoccer`, the robot is required to move a ball to the goal. For `cube` and `puzzle`, the agent manipulates a robotic arm with a 5-dimensional action space. The `cube` task requires pick-and-place, while the `puzzle` task involves pressing the buttons to solve the puzzle.

For D4RL, we evaluate on the following environments and datasets:

- antmaze-umaze-v2
- antmaze-umaze-diverse-v2
- antmaze-medium-play-v2
- antmaze-medium-diverse-v2
- antmaze-large-play-v2
- antmaze-large-diverse-v2
- pen-cloned-v1
- door-cloned-v1
- hammer-cloned-v1
- relocate-cloned-v1

The `antamze` tasks share the same 8-dimensional robot as in OGBench but differ in their environment and dataset settings. The `adroit` suite (`pen`, `door`, `hammer`, `relocate`) involves high-dimensional dexterous manipulation tasks, with action spaces exceeding 24 dimensions.

## F.2    EVALUATION

We evaluate all baselines during online fine-tuning by reporting the average return over 50 episodes every 50,000 environment steps. For OGBench and the antamze tasks in D4RL, we follow the original evaluation protocol and use the success rate as the performance metric, while for the adroit suite in D4RL, we adopt the normalized score. All experiments are conducted with 10 random seeds.

# G    HYPERPARAMETER SETTINGS

## G.1    FINO

Since our method builds on FQL (Park et al., 2025b) as the backbone algorithm, we retain all hyper-parameters from FQL without modification. The hyperparameters introduced in our method, namely $\alpha_t$ for noise injection and $N_{\text{sample}}$ and $\bar{\mathcal{H}}$ for entropy-guided sampling, are configured depending on the environment settings. The entropy update interval ($N_\xi$) is aligned with the evaluation frequency and set to 50,000 steps. A complete list of hyperparameters is provided in Table 2.

Table 2: Hyperparameters

| Hyperparameter | Value |
|---|---|
| Noise constant $\eta$ | $0.05 \cdot \|\mathcal{A}\|$ |
| Candidate action samples $N_{\text{sample}}$ | $0.5 \cdot \dim(\mathcal{A})$ |
| Target entropy $\bar{\mathcal{H}}$ | $-\dim(\mathcal{A})$ |
| Entropy update steps $N_\xi$ | 50,000 |
| Standard deviation of noise $\alpha_t$ | $\eta \cdot \exp(5(t-1))$ |

| Hyperparameter (from FQL) | Value |
|---|---|
| Learning rate | 0.0003 |
| Optimizer | Adam (Kingma & Ba, 2015) |
| Minibatch size | 256 |
| MLP dimensions | [512, 512, 512, 512] |
| Nonlinearity | GELU (Hendrycks & Gimpel, 2016) |
| Target network smoothing coefficient | 0.005 |
| Discount factor $\gamma$ | 0.99 (default), 0.995 (`antmaze-giant`, `humanoidmaze`, `antsoccer`) |
| Flow steps | 10 |
| Flow time sampling distribution | Unif([0, 1]) |
| Clipped double Q-learning | False (default), True (`adroit`, `antmaze-large`, `giant-navigate`) |
| BC coefficient $\alpha$ | Table 3 |

## G.2    OTHER BASELINES

For the other baselines, we retain the hyperparameters used in FQL (Park et al., 2025b). For Re-BRAC (Tarasov et al., 2023), we treat the actor bc coefficient ($\alpha_1$) and the critic bc coefficient ($\alpha_2$) as tunable hyperparameters, while keeping all other settings at their default values. For Cal-QL, the cql regularizer coefficient ($\alpha$) and the target action gap ($\beta$) are used as hyperparameters. Regarding the network size, we set it to [256, 256, 256, 256] for manipulation tasks and [512, 512, 512, 512] for locomotion tasks of OGBench, with all other parameters kept at their default values. For RLPD, we use a re-implementation of RLPD from the codebase of FQL and adopt the same configuration as FQL, including setting the update-to-data ratio to 1 and using two value functions. For IFQL, the only hyperparameter is the number of action samples ($N$). For FQL, the behavior cloning coefficient ($\alpha$) is the sole hyperparameter. The task-specific hyperparameters are summarized in Table 3.

Table 3: Task-specific hyperparameters for each baseline.

| Task | ReBRAC ($\alpha_1, \alpha_2$) | Cal-QL ($\alpha, \beta$) | IFQL ($N$) | FQL ($\alpha$) |
|---|---|---|---|---|
| humanoidmaze-medium-navigate-v0 | (0.01, 0.01) | (5, 0.8) | 32 | 100 |
| humanoidmaze-large-navigate-v0 | (0.01, 0.01) | (5, 0.8) | 32 | 30 |
| antmaze-large-navigate-v0 | (0.003, 0.01) | (5, 0.8) | 32 | 30 |
| antmaze-giant-navigate-v0 | (0.003, 0.01) | (5, 0.8) | 32 | 10 |
| antsoccer-arena-navigate-v0 | (0.01, 0.01) | (5, 0.2) | 64 | 30 |
| cube-double-play-v0 | (0.1, 0) | (0.01, 1) | 32 | 300 |
| puzzle-4x4-play-v0 | (0.3, 0.01) | (0.003, 1) | 32 | 1000 |
| antmaze-umaze-v2 | (0.003, 0.002) | (5, 0.8) | 32 | 10 |
| antmaze-umaze-diverse-v2 | (0.003, 0.001) | (5, 0.8) | 32 | 10 |
| antmaze-medium-play-v2 | (0.001, 0.0005) | (5, 0.8) | 32 | 10 |
| antmaze-medium-diverse-v2 | (0.001, 0) | (5, 0.8) | 32 | 10 |
| antmaze-large-play-v2 | (0.002, 0.001) | (5, 0.8) | 32 | 3 |
| antmaze-large-diverse-v2 | (0.002, 0.002) | (5, 0.8) | 32 | 3 |
| pen-cloned-v1 | (0.05, 0.5) | (1, 0.8) | 128 | 1000 |
| door-cloned-v1 | (0.01, 0.1) | (1, 0.8) | 128 | 1000 |
| hammer-cloned-v1 | (0.1, 0.5) | (1, 0.8) | 128 | 1000 |
| relocate-cloned-v1 | (0.1, 0.01) | (1, 0.8) | 128 | 10000 |

Table 4: Full results for main experiments (corresponding to Table 1 and Fig. 3). Scores show offline pre-training → online fine-tuning, averaged over 10 seeds (mean ± 95% CI). For OGBench, the `singletask` suffix is omitted.

| Environment | ReBRAC | Cal-QL | RLPD | IFQL | FQL | FINO |
|---|---|---|---|---|---|---|
| OGBench humanoidmaze-medium-navigate-task1 | $14_{\pm7} \to 1_{\pm1}$ | $0_{\pm0} \to 0_{\pm0}$ | $0_{\pm0} \to 0_{\pm0}$ | $65_{\pm18} \to 45_{\pm11}$ | $11_{\pm5} \to 14_{\pm4}$ | $13_{\pm3} \to \mathbf{91}_{\pm3}$ |
| OGBench humanoidmaze-medium-navigate-task2 | $18_{\pm9} \to 1_{\pm1}$ | $0_{\pm0} \to 0_{\pm0}$ | $0_{\pm0} \to 0_{\pm0}$ | $92_{\pm4} \to 83_{\pm7}$ | $89_{\pm13} \to 90_{\pm3}$ | $77_{\pm25} \to \mathbf{99}_{\pm1}$ |
| OGBench humanoidmaze-medium-navigate-task3 | $30_{\pm14} \to 1_{\pm1}$ | $0_{\pm0} \to 0_{\pm0}$ | $0_{\pm0} \to 1_{\pm1}$ | $38_{\pm30} \to 74_{\pm18}$ | $56_{\pm23} \to 89_{\pm2}$ | $52_{\pm24} \to \mathbf{99}_{\pm1}$ |
| OGBench humanoidmaze-medium-navigate-task4 | $17_{\pm11} \to 1_{\pm1}$ | $0_{\pm0} \to 0_{\pm0}$ | $0_{\pm0} \to 0_{\pm0}$ | $0_{\pm0} \to 56_{\pm9}$ | $9_{\pm12} \to 18_{\pm6}$ | $11_{\pm4} \to \mathbf{94}_{\pm3}$ |
| OGBench humanoidmaze-medium-navigate-task5 | $28_{\pm14} \to 10_{\pm16}$ | $0_{\pm0} \to 0_{\pm0}$ | $0_{\pm0} \to 4_{\pm3}$ | $99_{\pm1} \to 92_{\pm3}$ | $100_{\pm0} \to 94_{\pm3}$ | $99_{\pm1} \to \mathbf{100}_{\pm1}$ |
| OGBench humanoidmaze-large-navigate-task1 | $1_{\pm1} \to 0_{\pm0}$ | $0_{\pm0} \to 0_{\pm0}$ | $0_{\pm0} \to 0_{\pm0}$ | $1_{\pm1} \to 0_{\pm0}$ | $4_{\pm3} \to 1_{\pm1}$ | $5_{\pm4} \to \mathbf{5}_{\pm9}$ |
| OGBench humanoidmaze-large-navigate-task2 | $0_{\pm0} \to 0_{\pm0}$ | $0_{\pm0} \to 0_{\pm0}$ | $0_{\pm0} \to 0_{\pm0}$ | $0_{\pm0} \to 0_{\pm0}$ | $0_{\pm0} \to 0_{\pm0}$ | $0_{\pm0} \to \mathbf{6}_{\pm6}$ |
| OGBench humanoidmaze-large-navigate-task3 | $9_{\pm6} \to 2_{\pm2}$ | $0_{\pm0} \to 0_{\pm0}$ | $0_{\pm0} \to 0_{\pm0}$ | $45_{\pm10} \to 41_{\pm7}$ | $17_{\pm6} \to 36_{\pm11}$ | $22_{\pm10} \to \mathbf{99}_{\pm1}$ |
| OGBench humanoidmaze-large-navigate-task4 | $1_{\pm1} \to 1_{\pm1}$ | $0_{\pm0} \to 0_{\pm0}$ | $0_{\pm0} \to 0_{\pm0}$ | $0_{\pm0} \to 5_{\pm3}$ | $2_{\pm2} \to 8_{\pm8}$ | $0_{\pm0} \to \mathbf{48}_{\pm29}$ |
| OGBench humanoidmaze-large-navigate-task5 | $1_{\pm1} \to 0_{\pm1}$ | $0_{\pm0} \to 0_{\pm0}$ | $0_{\pm0} \to 0_{\pm0}$ | $8_{\pm10} \to 5_{\pm3}$ | $1_{\pm1} \to 5_{\pm7}$ | $0_{\pm0} \to \mathbf{8}_{\pm16}$ |
| OGBench antmaze-large-navigate-task1 | $94_{\pm4} \to \mathbf{100}_{\pm0}$ | $20_{\pm26} \to 30_{\pm30}$ | $0_{\pm0} \to 93_{\pm8}$ | $32_{\pm10} \to 66_{\pm15}$ | $82_{\pm5} \to 98_{\pm1}$ | $82_{\pm6} \to 98_{\pm2}$ |
| OGBench antmaze-large-navigate-task2 | $88_{\pm2} \to \mathbf{98}_{\pm1}$ | $0_{\pm0} \to 0_{\pm0}$ | $0_{\pm0} \to 54_{\pm25}$ | $17_{\pm7} \to 70_{\pm4}$ | $63_{\pm5} \to 71_{\pm4}$ | $62_{\pm6} \to 97_{\pm1}$ |
| OGBench antmaze-large-navigate-task3 | $65_{\pm14} \to \mathbf{100}_{\pm0}$ | $20_{\pm26} \to 10_{\pm20}$ | $0_{\pm0} \to 99_{\pm1}$ | $57_{\pm9} \to 88_{\pm4}$ | $94_{\pm2} \to \mathbf{100}_{\pm1}$ | $92_{\pm3} \to \mathbf{100}_{\pm0}$ |
| OGBench antmaze-large-navigate-task4 | $86_{\pm5} \to \mathbf{99}_{\pm1}$ | $0_{\pm0} \to 0_{\pm0}$ | $0_{\pm0} \to 86_{\pm11}$ | $13_{\pm5} \to 75_{\pm8}$ | $80_{\pm5} \to 96_{\pm1}$ | $83_{\pm4} \to \mathbf{99}_{\pm1}$ |
| OGBench antmaze-large-navigate-task5 | $90_{\pm4} \to \mathbf{100}_{\pm1}$ | $20_{\pm26} \to 20_{\pm26}$ | $0_{\pm0} \to 69_{\pm24}$ | $39_{\pm10} \to 59_{\pm23}$ | $85_{\pm4} \to 95_{\pm2}$ | $85_{\pm5} \to 99_{\pm1}$ |
| OGBench antmaze-giant-navigate-task1 | $53_{\pm16} \to \mathbf{97}_{\pm1}$ | $0_{\pm0} \to 0_{\pm0}$ | $0_{\pm0} \to 7_{\pm12}$ | $0_{\pm0} \to 0_{\pm0}$ | $8_{\pm7} \to 65_{\pm24}$ | $3_{\pm4} \to \mathbf{96}_{\pm1}$ |
| OGBench antmaze-giant-navigate-task2 | $25_{\pm17} \to \mathbf{98}_{\pm1}$ | $0_{\pm0} \to 0_{\pm0}$ | $0_{\pm0} \to 48_{\pm24}$ | $0_{\pm0} \to 0_{\pm0}$ | $17_{\pm11} \to 96_{\pm1}$ | $0_{\pm1} \to \mathbf{99}_{\pm1}$ |
| OGBench antmaze-giant-navigate-task3 | $34_{\pm20} \to \mathbf{86}_{\pm19}$ | $0_{\pm0} \to 0_{\pm0}$ | $0_{\pm0} \to 44_{\pm18}$ | $0_{\pm0} \to 0_{\pm0}$ | $0_{\pm1} \to 2_{\pm2}$ | $0_{\pm0} \to 0_{\pm0}$ |
| OGBench antmaze-giant-navigate-task4 | $0_{\pm0} \to \mathbf{98}_{\pm1}$ | $0_{\pm0} \to 0_{\pm0}$ | $0_{\pm0} \to 59_{\pm26}$ | $0_{\pm0} \to 0_{\pm0}$ | $10_{\pm13} \to \mathbf{96}_{\pm1}$ | $29_{\pm23} \to \mathbf{99}_{\pm1}$ |
| OGBench antmaze-giant-navigate-task5 | $61_{\pm12} \to \mathbf{99}_{\pm1}$ | $10_{\pm20} \to 0_{\pm0}$ | $0_{\pm0} \to 80_{\pm12}$ | $4_{\pm4} \to 0_{\pm0}$ | $43_{\pm21} \to \mathbf{99}_{\pm1}$ | $36_{\pm15} \to \mathbf{99}_{\pm1}$ |
| OGBench antsoccer-arena-navigate-task1 | $1_{\pm1} \to 0_{\pm0}$ | $0_{\pm0} \to 0_{\pm0}$ | $0_{\pm0} \to 6_{\pm4}$ | $69_{\pm15} \to 64_{\pm10}$ | $82_{\pm4} \to 91_{\pm2}$ | $77_{\pm6} \to \mathbf{93}_{\pm2}$ |
| OGBench antsoccer-arena-navigate-task2 | $0_{\pm1} \to 0_{\pm0}$ | $0_{\pm0} \to 0_{\pm0}$ | $0_{\pm0} \to 5_{\pm4}$ | $70_{\pm7} \to 66_{\pm21}$ | $88_{\pm4} \to 97_{\pm3}$ | $84_{\pm5} \to \mathbf{98}_{\pm1}$ |
| OGBench antsoccer-arena-navigate-task3 | $0_{\pm0} \to 0_{\pm0}$ | $0_{\pm0} \to 0_{\pm0}$ | $0_{\pm0} \to 1_{\pm1}$ | $6_{\pm6} \to 26_{\pm9}$ | $60_{\pm4} \to \mathbf{88}_{\pm4}$ | $56_{\pm5} \to 91_{\pm2}$ |
| OGBench antsoccer-arena-navigate-task4 | $1_{\pm1} \to 0_{\pm0}$ | $0_{\pm0} \to 0_{\pm0}$ | $0_{\pm0} \to 0_{\pm0}$ | $20_{\pm9} \to 17_{\pm6}$ | $32_{\pm4} \to \mathbf{70}_{\pm5}$ | $34_{\pm6} \to \mathbf{70}_{\pm6}$ |
| OGBench antsoccer-arena-navigate-task5 | $0_{\pm0} \to 0_{\pm0}$ | $0_{\pm0} \to 0_{\pm0}$ | $0_{\pm0} \to 0_{\pm0}$ | $1_{\pm2} \to 4_{\pm3}$ | $43_{\pm9} \to 22_{\pm18}$ | $32_{\pm7} \to \mathbf{33}_{\pm25}$ |
| OGBench cube-double-play-task1 | $27_{\pm8} \to \mathbf{100}_{\pm0}$ | $10_{\pm20} \to 0_{\pm0}$ | $0_{\pm0} \to 11_{\pm15}$ | $31_{\pm5} \to 89_{\pm3}$ | $59_{\pm9} \to 97_{\pm2}$ | $62_{\pm5} \to 98_{\pm1}$ |
| OGBench cube-double-play-task2 | $8_{\pm3} \to 20_{\pm6}$ | $0_{\pm0} \to 0_{\pm0}$ | $0_{\pm0} \to 0_{\pm0}$ | $13_{\pm3} \to 29_{\pm4}$ | $42_{\pm9} \to 86_{\pm7}$ | $40_{\pm7} \to \mathbf{90}_{\pm3}$ |
| OGBench cube-double-play-task3 | $3_{\pm2} \to 16_{\pm7}$ | $0_{\pm0} \to 0_{\pm0}$ | $0_{\pm0} \to 0_{\pm0}$ | $6_{\pm2} \to 31_{\pm7}$ | $29_{\pm5} \to 89_{\pm9}$ | $35_{\pm8} \to \mathbf{90}_{\pm4}$ |
| OGBench cube-double-play-task4 | $1_{\pm1} \to 0_{\pm1}$ | $0_{\pm0} \to 0_{\pm0}$ | $0_{\pm0} \to 0_{\pm0}$ | $1_{\pm1} \to 0_{\pm0}$ | $5_{\pm3} \to 4_{\pm2}$ | $11_{\pm4} \to \mathbf{21}_{\pm8}$ |
| OGBench cube-double-play-task5 | $3_{\pm2} \to 3_{\pm2}$ | $0_{\pm0} \to 0_{\pm0}$ | $0_{\pm0} \to 0_{\pm0}$ | $17_{\pm3} \to 51_{\pm6}$ | $20_{\pm6} \to 88_{\pm3}$ | $23_{\pm10} \to \mathbf{96}_{\pm3}$ |
| OGBench puzzle-4x4-play-task1 | $25_{\pm4} \to \mathbf{100}_{\pm0}$ | $10_{\pm20} \to 70_{\pm30}$ | $0_{\pm0} \to \mathbf{100}_{\pm0}$ | $38_{\pm6} \to \mathbf{100}_{\pm1}$ | $32_{\pm6} \to \mathbf{100}_{\pm0}$ | $33_{\pm8} \to \mathbf{100}_{\pm0}$ |
| OGBench puzzle-4x4-play-task2 | $10_{\pm4} \to 0_{\pm0}$ | $0_{\pm0} \to 0_{\pm0}$ | $0_{\pm0} \to \mathbf{40}_{\pm32}$ | $16_{\pm5} \to 1_{\pm1}$ | $13_{\pm4} \to 0_{\pm0}$ | $11_{\pm4} \to 0_{\pm0}$ |
| OGBench puzzle-4x4-play-task3 | $16_{\pm4} \to 42_{\pm26}$ | $0_{\pm0} \to 30_{\pm30}$ | $0_{\pm0} \to 89_{\pm19}$ | $49_{\pm8} \to 91_{\pm12}$ | $18_{\pm5} \to 70_{\pm28}$ | $22_{\pm9} \to \mathbf{100}_{\pm0}$ |
| OGBench puzzle-4x4-play-task4 | $8_{\pm2} \to 3_{\pm3}$ | $0_{\pm0} \to 0_{\pm0}$ | $0_{\pm0} \to 40_{\pm32}$ | $20_{\pm4} \to 19_{\pm17}$ | $7_{\pm4} \to 53_{\pm30}$ | $20_{\pm5} \to \mathbf{80}_{\pm24}$ |
| OGBench puzzle-4x4-play-task5 | $9_{\pm2} \to 0_{\pm0}$ | $0_{\pm0} \to 0_{\pm0}$ | $0_{\pm0} \to \mathbf{20}_{\pm26}$ | $8_{\pm3} \to 0_{\pm0}$ | $5_{\pm2} \to 0_{\pm0}$ | $8_{\pm2} \to 0_{\pm0}$ |
| D4RL antmaze-umaze-v2 | $90_{\pm4} \to \mathbf{100}_{\pm1}$ | $81_{\pm3} \to 99_{\pm1}$ | $0_{\pm0} \to \mathbf{100}_{\pm1}$ | $94_{\pm2} \to 96_{\pm2}$ | $98_{\pm1} \to 99_{\pm1}$ | $98_{\pm2} \to \mathbf{100}_{\pm1}$ |
| D4RL antmaze-umaze-diverse-v2 | $75_{\pm12} \to \mathbf{100}_{\pm1}$ | $36_{\pm12} \to 94_{\pm4}$ | $0_{\pm0} \to 99_{\pm1}$ | $71_{\pm14} \to 53_{\pm22}$ | $85_{\pm7} \to 99_{\pm1}$ | $78_{\pm5} \to 99_{\pm1}$ |
| D4RL antmaze-medium-play-v2 | $8_{\pm8} \to 91_{\pm11}$ | $60_{\pm12} \to 91_{\pm11}$ | $0_{\pm0} \to \mathbf{97}_{\pm1}$ | $54_{\pm14} \to 79_{\pm18}$ | $78_{\pm5} \to 94_{\pm2}$ | $79_{\pm5} \to \mathbf{97}_{\pm1}$ |
| D4RL antmaze-medium-diverse-v2 | $11_{\pm8} \to \mathbf{98}_{\pm2}$ | $61_{\pm5} \to 95_{\pm2}$ | $0_{\pm0} \to \mathbf{98}_{\pm1}$ | $41_{\pm20} \to 86_{\pm3}$ | $66_{\pm9} \to 95_{\pm2}$ | $60_{\pm11} \to 97_{\pm1}$ |
| D4RL antmaze-large-play-v2 | $42_{\pm21} \to 51_{\pm28}$ | $35_{\pm5} \to 74_{\pm6}$ | $0_{\pm0} \to 79_{\pm11}$ | $64_{\pm5} \to 78_{\pm3}$ | $73_{\pm18} \to \mathbf{95}_{\pm1}$ | $75_{\pm17} \to \mathbf{95}_{\pm1}$ |
| D4RL antmaze-large-diverse-v2 | $77_{\pm5} \to \mathbf{93}_{\pm2}$ | $29_{\pm8} \to 80_{\pm4}$ | $0_{\pm0} \to 83_{\pm8}$ | $73_{\pm6} \to 84_{\pm3}$ | $81_{\pm11} \to 91_{\pm2}$ | $82_{\pm4} \to \mathbf{97}_{\pm1}$ |
| D4RL pen-cloned-v1 | $77_{\pm6} \to 129_{\pm4}$ | $-1_{\pm1} \to -1_{\pm1}$ | $3_{\pm2} \to 93_{\pm8}$ | $73_{\pm3} \to 97_{\pm7}$ | $53_{\pm12} \to \mathbf{141}_{\pm4}$ | $51_{\pm8} \to \mathbf{140}_{\pm3}$ |
| D4RL door-cloned-v1 | $0_{\pm0} \to 83_{\pm6}$ | $0_{\pm0} \to 0_{\pm0}$ | $0_{\pm0} \to \mathbf{101}_{\pm4}$ | $1_{\pm1} \to 23_{\pm5}$ | $0_{\pm0} \to \mathbf{101}_{\pm2}$ | $0_{\pm0} \to \mathbf{104}_{\pm1}$ |
| D4RL hammer-cloned-v1 | $5_{\pm3} \to 119_{\pm2}$ | $0_{\pm0} \to 0_{\pm0}$ | $0_{\pm0} \to 99_{\pm19}$ | $1_{\pm1} \to 44_{\pm8}$ | $0_{\pm0} \to 110_{\pm25}$ | $0_{\pm0} \to \mathbf{134}_{\pm2}$ |
| D4RL relocate-cloned-v1 | $0_{\pm0} \to 0_{\pm0}$ | $0_{\pm0} \to 0_{\pm0}$ | $0_{\pm0} \to 0_{\pm0}$ | $0_{\pm0} \to 2_{\pm1}$ | $1_{\pm0} \to 47_{\pm2}$ | $1_{\pm0} \to \mathbf{72}_{\pm2}$ |

Table 5: Full results for ablation studies (Fig. 5, Fig. 6). Scores show offline pre-training $\to$ online fine-tuning, averaged over 10 seeds (mean $\pm$ 95% CI). For OGBench, the `singletask` suffix is omitted.

| Environment | FINO | Direct Noise | w/o Noise | w/o Guidance |
|---|---|---|---|---|
| OGBench humanoidmaze-medium-navigate-task1 | $13_{\pm3} \to \mathbf{91}_{\pm3}$ | $14_{\pm5} \to 0_{\pm1}$ | $18_{\pm5} \to 0_{\pm0}$ | $16_{\pm3} \to 50_{\pm16}$ |
| OGBench humanoidmaze-medium-navigate-task2 | $77_{\pm25} \to \mathbf{99}_{\pm1}$ | $88_{\pm16} \to \mathbf{96}_{\pm6}$ | $80_{\pm19} \to \mathbf{99}_{\pm2}$ | $71_{\pm25} \to 84_{\pm20}$ |
| OGBench humanoidmaze-medium-navigate-task3 | $52_{\pm24} \to \mathbf{99}_{\pm1}$ | $54_{\pm22} \to 60_{\pm32}$ | $45_{\pm24} \to 49_{\pm32}$ | $48_{\pm23} \to 79_{\pm25}$ |
| OGBench humanoidmaze-medium-navigate-task4 | $11_{\pm4} \to \mathbf{94}_{\pm3}$ | $7_{\pm10} \to 31_{\pm24}$ | $2_{\pm4} \to 20_{\pm26}$ | $0_{\pm0} \to 74_{\pm16}$ |
| OGBench humanoidmaze-medium-navigate-task5 | $99_{\pm1} \to \mathbf{100}_{\pm1}$ | $99_{\pm1} \to \mathbf{100}_{\pm1}$ | $99_{\pm1} \to \mathbf{99}_{\pm1}$ | $98_{\pm1} \to \mathbf{99}_{\pm1}$ |
| OGBench humanoidmaze-large-navigate-task1 | $5_{\pm4} \to 5_{\pm9}$ | $4_{\pm4} \to 0_{\pm0}$ | $3_{\pm4} \to \mathbf{8}_{\pm14}$ | $4_{\pm4} \to 0_{\pm1}$ |
| OGBench humanoidmaze-large-navigate-task2 | $0_{\pm0} \to \mathbf{6}_{\pm6}$ | $0_{\pm0} \to 0_{\pm0}$ | $0_{\pm0} \to 2_{\pm3}$ | $0_{\pm1} \to 3_{\pm5}$ |
| OGBench humanoidmaze-large-navigate-task3 | $22_{\pm10} \to \mathbf{99}_{\pm1}$ | $19_{\pm6} \to 41_{\pm5}$ | $18_{\pm7} \to \mathbf{95}_{\pm4}$ | $25_{\pm10} \to 78_{\pm26}$ |
| OGBench humanoidmaze-large-navigate-task4 | $0_{\pm0} \to \mathbf{48}_{\pm29}$ | $2_{\pm2} \to 8_{\pm7}$ | $0_{\pm0} \to 5_{\pm9}$ | $0_{\pm0} \to 18_{\pm22}$ |
| OGBench humanoidmaze-large-navigate-task5 | $0_{\pm0} \to 8_{\pm16}$ | $0_{\pm0} \to 4_{\pm4}$ | $1_{\pm2} \to 0_{\pm0}$ | $0_{\pm0} \to \mathbf{13}_{\pm18}$ |
| OGBench antmaze-large-navigate-task1 | $82_{\pm6} \to \mathbf{98}_{\pm2}$ | $82_{\pm5} \to \mathbf{98}_{\pm1}$ | $68_{\pm15} \to \mathbf{98}_{\pm1}$ | $78_{\pm5} \to 35_{\pm27}$ |
| OGBench antmaze-large-navigate-task2 | $62_{\pm6} \to \mathbf{97}_{\pm1}$ | $59_{\pm5} \to 69_{\pm8}$ | $62_{\pm5} \to \mathbf{94}_{\pm3}$ | $75_{\pm6} \to 91_{\pm3}$ |
| OGBench antmaze-large-navigate-task3 | $92_{\pm3} \to \mathbf{100}_{\pm0}$ | $96_{\pm2} \to \mathbf{99}_{\pm1}$ | $95_{\pm2} \to \mathbf{100}_{\pm0}$ | $95_{\pm3} \to \mathbf{99}_{\pm1}$ |
| OGBench antmaze-large-navigate-task4 | $83_{\pm4} \to \mathbf{99}_{\pm1}$ | $72_{\pm16} \to \mathbf{96}_{\pm2}$ | $69_{\pm15} \to \mathbf{98}_{\pm1}$ | $81_{\pm4} \to \mathbf{98}_{\pm1}$ |
| OGBench antmaze-large-navigate-task5 | $85_{\pm5} \to \mathbf{99}_{\pm1}$ | $83_{\pm3} \to \mathbf{95}_{\pm2}$ | $83_{\pm3} \to \mathbf{98}_{\pm2}$ | $82_{\pm5} \to 83_{\pm21}$ |
| OGBench antmaze-giant-navigate-task1 | $3_{\pm4} \to \mathbf{96}_{\pm1}$ | $9_{\pm7} \to 64_{\pm22}$ | $9_{\pm10} \to 85_{\pm13}$ | $4_{\pm5} \to 79_{\pm26}$ |
| OGBench antmaze-giant-navigate-task2 | $0_{\pm1} \to \mathbf{99}_{\pm1}$ | $18_{\pm11} \to \mathbf{96}_{\pm1}$ | $1_{\pm2} \to \mathbf{98}_{\pm2}$ | $1_{\pm2} \to \mathbf{99}_{\pm1}$ |
| OGBench antmaze-giant-navigate-task3 | $0_{\pm0} \to 0_{\pm0}$ | $0_{\pm1} \to 1_{\pm2}$ | $0_{\pm0} \to 0_{\pm0}$ | $0_{\pm0} \to \mathbf{7}_{\pm15}$ |
| OGBench antmaze-giant-navigate-task4 | $29_{\pm23} \to \mathbf{99}_{\pm1}$ | $10_{\pm12} \to 85_{\pm15}$ | $5_{\pm7} \to \mathbf{98}_{\pm1}$ | $28_{\pm23} \to \mathbf{97}_{\pm3}$ |
| OGBench antmaze-giant-navigate-task5 | $36_{\pm15} \to \mathbf{99}_{\pm1}$ | $43_{\pm20} \to \mathbf{98}_{\pm1}$ | $22_{\pm13} \to \mathbf{99}_{\pm1}$ | $31_{\pm17} \to \mathbf{99}_{\pm1}$ |
| OGBench antsoccer-arena-navigate-task1 | $77_{\pm6} \to \mathbf{93}_{\pm2}$ | $81_{\pm4} \to \mathbf{93}_{\pm4}$ | $69_{\pm7} \to \mathbf{94}_{\pm2}$ | $68_{\pm5} \to 73_{\pm16}$ |
| OGBench antsoccer-arena-navigate-task2 | $84_{\pm5} \to \mathbf{98}_{\pm1}$ | $90_{\pm3} \to \mathbf{97}_{\pm2}$ | $83_{\pm6} \to \mathbf{97}_{\pm2}$ | $83_{\pm4} \to 93_{\pm2}$ |
| OGBench antsoccer-arena-navigate-task3 | $56_{\pm5} \to \mathbf{91}_{\pm2}$ | $58_{\pm4} \to \mathbf{87}_{\pm4}$ | $54_{\pm4} \to 81_{\pm4}$ | $57_{\pm4} \to 78_{\pm4}$ |
| OGBench antsoccer-arena-navigate-task4 | $34_{\pm6} \to \mathbf{70}_{\pm6}$ | $33_{\pm4} \to \mathbf{71}_{\pm8}$ | $43_{\pm5} \to 62_{\pm7}$ | $43_{\pm4} \to 50_{\pm11}$ |
| OGBench antsoccer-arena-navigate-task5 | $32_{\pm7} \to 33_{\pm25}$ | $43_{\pm7} \to 14_{\pm17}$ | $19_{\pm5} \to 10_{\pm19}$ | $33_{\pm8} \to \mathbf{42}_{\pm23}$ |
| OGBench cube-double-play-task1 | $62_{\pm5} \to \mathbf{98}_{\pm1}$ | $64_{\pm9} \to \mathbf{97}_{\pm3}$ | $73_{\pm6} \to \mathbf{95}_{\pm3}$ | $74_{\pm8} \to \mathbf{96}_{\pm3}$ |
| OGBench cube-double-play-task2 | $40_{\pm7} \to \mathbf{90}_{\pm3}$ | $40_{\pm5} \to 86_{\pm3}$ | $60_{\pm7} \to 80_{\pm9}$ | $61_{\pm9} \to \mathbf{89}_{\pm5}$ |
| OGBench cube-double-play-task3 | $35_{\pm8} \to 90_{\pm4}$ | $26_{\pm5} \to 88_{\pm6}$ | $57_{\pm5} \to 88_{\pm5}$ | $52_{\pm7} \to \mathbf{96}_{\pm2}$ |
| OGBench cube-double-play-task4 | $11_{\pm4} \to \mathbf{21}_{\pm8}$ | $5_{\pm2} \to 3_{\pm2}$ | $14_{\pm1} \to 2_{\pm1}$ | $8_{\pm3} \to 6_{\pm4}$ |
| OGBench cube-double-play-task5 | $23_{\pm10} \to \mathbf{96}_{\pm3}$ | $21_{\pm6} \to 88_{\pm5}$ | $43_{\pm15} \to 86_{\pm6}$ | $26_{\pm9} \to 91_{\pm4}$ |
| OGBench puzzle-4x4-play-task1 | $33_{\pm8} \to \mathbf{100}_{\pm0}$ | $31_{\pm5} \to \mathbf{100}_{\pm0}$ | $56_{\pm7} \to \mathbf{100}_{\pm0}$ | $61_{\pm7} \to \mathbf{100}_{\pm0}$ |
| OGBench puzzle-4x4-play-task2 | $11_{\pm4} \to 0_{\pm0}$ | $12_{\pm2} \to 0_{\pm0}$ | $15_{\pm5} \to 0_{\pm0}$ | $10_{\pm3} \to 0_{\pm0}$ |
| OGBench puzzle-4x4-play-task3 | $22_{\pm9} \to \mathbf{100}_{\pm0}$ | $20_{\pm2} \to 80_{\pm26}$ | $53_{\pm11} \to 88_{\pm15}$ | $59_{\pm6} \to \mathbf{97}_{\pm6}$ |
| OGBench puzzle-4x4-play-task4 | $20_{\pm5} \to \mathbf{80}_{\pm24}$ | $9_{\pm3} \to 61_{\pm31}$ | $18_{\pm4} \to 33_{\pm29}$ | $19_{\pm6} \to 10_{\pm20}$ |
| OGBench puzzle-4x4-play-task5 | $8_{\pm2} \to 0_{\pm0}$ | $7_{\pm3} \to 0_{\pm0}$ | $6_{\pm3} \to 0_{\pm0}$ | $9_{\pm4} \to 0_{\pm0}$ |
| D4RL antmaze-umaze-v2 | $98_{\pm2} \to \mathbf{100}_{\pm1}$ | $96_{\pm2} \to \mathbf{99}_{\pm1}$ | $98_{\pm1} \to \mathbf{99}_{\pm1}$ | $97_{\pm2} \to \mathbf{99}_{\pm1}$ |
| D4RL antmaze-umaze-diverse-v2 | $78_{\pm5} \to \mathbf{99}_{\pm1}$ | $85_{\pm5} \to \mathbf{97}_{\pm1}$ | $85_{\pm7} \to \mathbf{99}_{\pm1}$ | $82_{\pm6} \to \mathbf{100}_{\pm0}$ |
| D4RL antmaze-medium-play-v2 | $79_{\pm2} \to \mathbf{97}_{\pm1}$ | $74_{\pm4} \to 93_{\pm3}$ | $80_{\pm5} \to \mathbf{96}_{\pm2}$ | $79_{\pm4} \to \mathbf{96}_{\pm2}$ |
| D4RL antmaze-medium-diverse-v2 | $60_{\pm11} \to \mathbf{97}_{\pm1}$ | $62_{\pm8} \to \mathbf{95}_{\pm1}$ | $62_{\pm10} \to \mathbf{95}_{\pm6}$ | $61_{\pm11} \to \mathbf{97}_{\pm1}$ |
| D4RL antmaze-large-play-v2 | $75_{\pm17} \to \mathbf{95}_{\pm1}$ | $72_{\pm19} \to 90_{\pm5}$ | $67_{\pm22} \to \mathbf{92}_{\pm2}$ | $73_{\pm16} \to \mathbf{94}_{\pm3}$ |
| D4RL antmaze-large-diverse-v2 | $82_{\pm4} \to \mathbf{97}_{\pm1}$ | $83_{\pm9} \to 90_{\pm4}$ | $72_{\pm16} \to \mathbf{94}_{\pm3}$ | $81_{\pm3} \to \mathbf{94}_{\pm3}$ |
| D4RL pen-cloned-v1 | $51_{\pm8} \to \mathbf{140}_{\pm3}$ | $57_{\pm9} \to \mathbf{137}_{\pm4}$ | $60_{\pm8} \to \mathbf{135}_{\pm4}$ | $57_{\pm7} \to \mathbf{136}_{\pm4}$ |
| D4RL door-cloned-v1 | $0_{\pm0} \to \mathbf{104}_{\pm1}$ | $0_{\pm0} \to \mathbf{102}_{\pm2}$ | $0_{\pm0} \to \mathbf{102}_{\pm2}$ | $0_{\pm0} \to \mathbf{101}_{\pm4}$ |
| D4RL hammer-cloned-v1 | $0_{\pm0} \to \mathbf{134}_{\pm2}$ | $0_{\pm0} \to 116_{\pm26}$ | $0_{\pm0} \to 120_{\pm12}$ | $0_{\pm0} \to 112_{\pm14}$ |
| D4RL relocate-cloned-v1 | $1_{\pm0} \to \mathbf{72}_{\pm2}$ | $1_{\pm0} \to 59_{\pm3}$ | $1_{\pm0} \to 61_{\pm5}$ | $1_{\pm1} \to 62_{\pm7}$ |

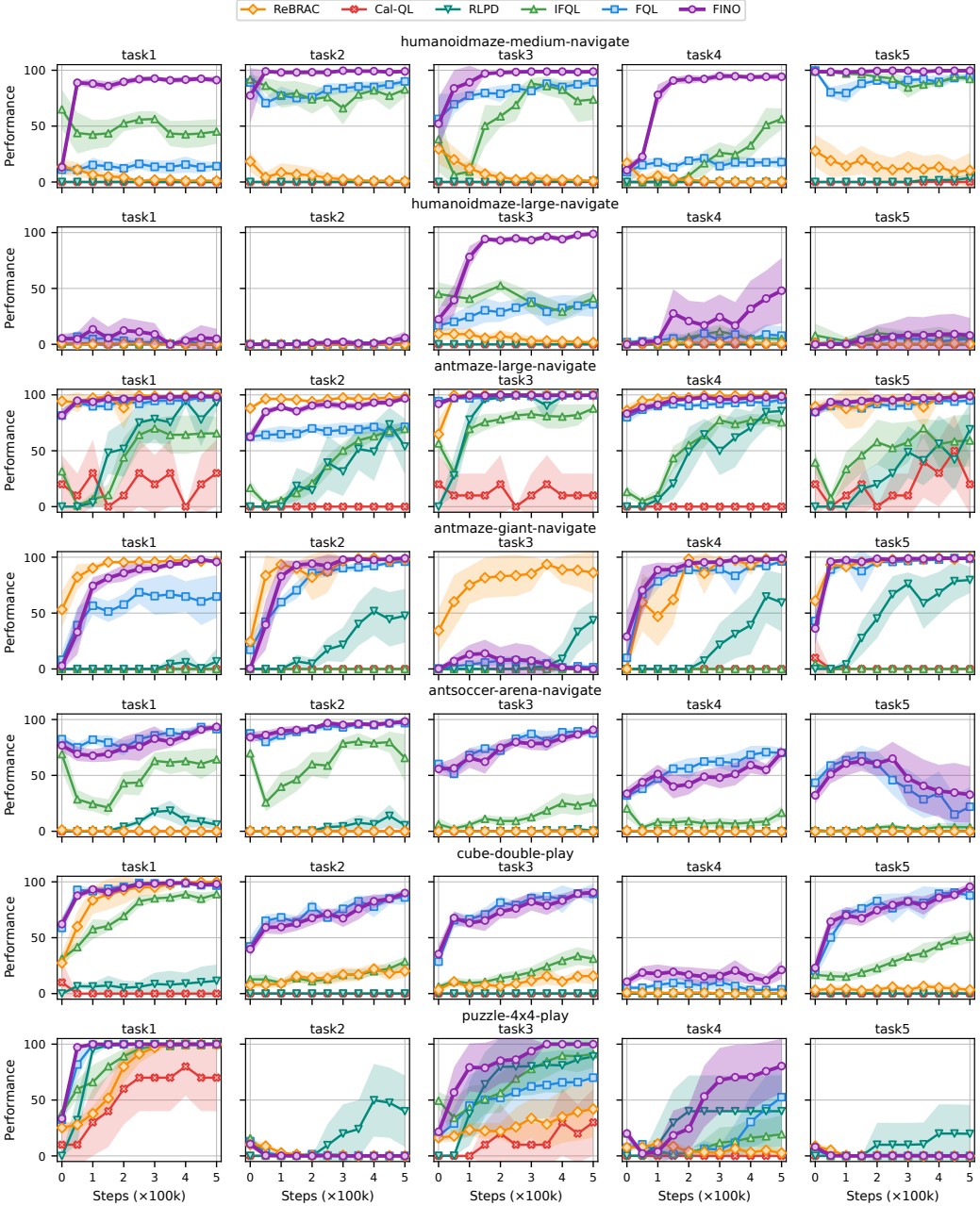

Figure 9: Full results on OGBench environments. Each row corresponds to one environment, with five single-task variants shown side by side. Shaded areas denote 95% confidence intervals over 10 seeds.

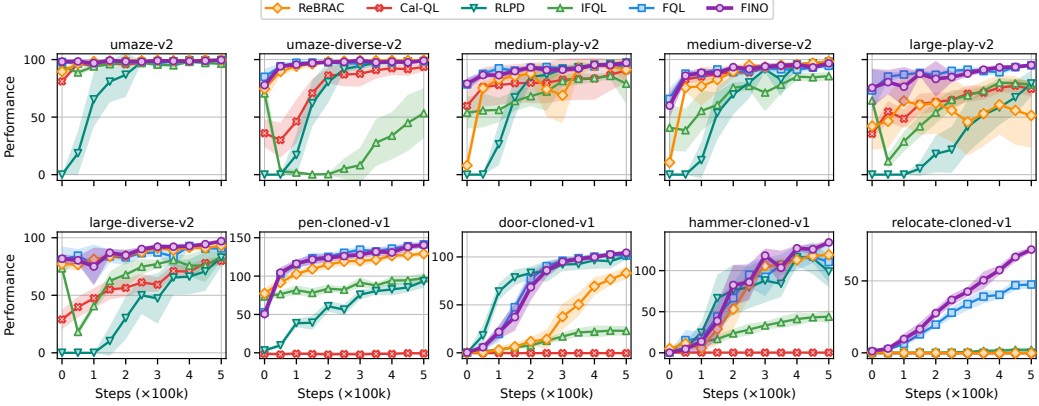

Figure 10: Full results on D4RL environments. For AntMaze tasks, the prefix "antmaze-" is omitted for clarity. Shaded areas denote 95% confidence intervals over 10 seeds.

