# OpenReview forum: "Flow Matching with Injected Noise for Offline-to-Online Reinforcement Learning"
_ICLR.cc/2026/Conference — ICLR 2026 Poster_

### Official Review · Reviewer_QgFC · 2025-11-01

**Soundness:** 3
**Presentation:** 3
**Contribution:** 3
**Rating:** 8
**Confidence:** 3

**Summary:**

This paper introduces a new approach for offline-to-online RL via injecting noise into the flow matching objective for improved exploration.

**Strengths:**

To my knowledge, this is the first paper to introduce the idea of noise injection into flow matching for offline-to-online RL, addressing a key limitation of prior work in offline-to-online RL. The empirical results demonstrate strong performance over the existing baselines (Tables 1 and 4), and attempts to justify the benefit of their method through ablations, though I provide comments about the ablations below.

**Weaknesses:**

The weaknesses I identify can be grouped broadly into three categories: exposition/justification of FINO’s objective, baselines and ablations, and smaller questions.

I believe greater explanation and exposition of the noise injection for flow matching objective is needed. For example, why is this objective better than simply adding noise to the velocity target, $x_1 - x_0$? Additionally, is $\eta$ fixed throughout training?

Do any of the baselines considered employ a similar policy extraction scheme, specifically a Q-value weighted softmax? The authors argue that directly injecting noise into action selection is less effective than FINO (Figure 4), but the direct noise injection seemed to happen only at inference time.

A relevant baseline may be adding noise to the offline dataset’s actions during training, in addition to at inference-time in the online stage.

The authors do not clearly define what type of noise is added in the Direct Noise ablation (Figure 4), only referring to it as “random perturbations.” I would suggest the authors investigate adding Gaussian noise to the actions, with varying standard deviations, similar to what they do for FINO.

**Questions:**

What does the y-axis scale on the right-hand side of Figure 2 represent? Also see questions above.

---

> ### Author Response · Authors · 2025-11-21
> **Authors' response to reviewer QgFC**
>
> We thank the reviewer for the insightful and constructive feedback.
> The suggestions help us refine the clarity and rigor of our work. Please note that the current PDF file at openreview is the revised paper file.
>
> Our responses to each point are outlined below.
>
> &nbsp;
>
> ### **R3-1. Regarding Noise Injection in Flow Matching**
> We appreciate the reviewer for raising this insightful point.
> When considering the noise injection mechanism, two straightforward alternatives naturally arise.
> The first is to add noise to the velocity target $x_1-x_0$, as the reviewer suggested.
> The second is to inject noise directly into the action output of the policy.
>
> In the first case, adding noise to the velocity target in Equation 4 yields:
>
> $$
> \\begin{aligned}
> &\\mathbb{E}[\\|v _{\\theta}(t,s,x _t)-(x _1-x _0)-\\epsilon\\| _2^2] \\\\
> =&\\mathbb{E}[\\|v _{\\theta} (t,s,x _t)-(x _1-x _0)\\| _2^2 -2 (v _{\\theta}(t,s,x _t)-(x _1 - x _0))^\\top \\epsilon + \\|\epsilon\\|_2^2]
> \\end{aligned}$$
>
> Since the noise has zero mean, the second term vanishes in expectation, and the final term is constant with respect to $\\theta$.
> As a result, the injected noise does not influence the gradient, and the optimization is identical to the original objective.
> This means that the flow model cannot broaden its learned action space, failing to meet our intended goal.
>
> The second approach injects noise directly into the action output of the policy.
> In this case, the behavior of the one-step policy that interacts with the environment differs substantially.
> As shown in Equation 5, the one-step policy is learned by distillation from the flow model and action-value maximization via the value function.
> Whereas original training maximizes action-value only within the dataset distribution, FINO benefits from a flow model that is trained over a wider region of the action space.
> When combined with action-value maximization, this enables the one-step policy to explore toward higher-value regions, rather than applying undirected random perturbations.
>
> In Appendix D of the revised manuscript, we provided a concrete example showing the clear difference between FINO and simply adding noise to actions.
> The results (Figure 8) show that the one-step policy of FINO consistently explores toward more promising directions, guiding more effective online fine-tuning than the baseline simply adding noise to actions.
> To further aid understanding of our method, we also incorporated the explanation of this mechanism into Section 4.1 of the revised manuscript.
>
> &nbsp;
>
> ### **R3-2. Clarification and Extension of Direct Noise Baselines**
> To clarify and strengthen the baselines, we add a new baseline in Section 6.1 and refine the explanation of the baseline in Section 6.2 in the revised manuscript.
>
> In Section 6.1, we compare FINO with a baseline that injects noise directly into the action generated by the policy (denoted as Action Noise), rather than injecting noise into the flow matching objective as in our method.
> The injected noise is Gaussian with the same mean and variance used in FINO and is applied only during training; no noise is added at inference time to ensure accurate evaluation.
> The entropy-guided sampling is kept identical to FINO.
> The results show that our proposed noise injection strategy is effective across different tasks and outperforms the direct action noise baseline.
>
> Section 6.2 (formerly the Direct Noise baseline) also injects Gaussian noise directly into the actions, as in Section 6.1.
> However, instead of using a fixed variance, this baseline modulates the Gaussian noise variance based on the policy’s entropy.
> To avoid confusion with the baseline in Section 6.1, we renamed this baseline to ER-Noise (Entropy Regulated Noise) and refined its description accordingly.
> The results indicate that simply modulating noise variance does not yield strong performance.
>
> Together, these findings show that directly injecting noise into actions is insufficient, and that injecting noise through the learning objective provides a more effective approach for online fine-tuning.
>
> &nbsp;
>
> ### **R3-3. Regarding Policy Extraction Mechanism**
> The revised manuscript includes two baselines that also use a Q-value weighted softmax for policy extraction.
> Both the Action Noise baseline in Figure 4 and the w/o Noise baseline in Figure 6 employ exactly the same policy extraction mechanism as FINO.
> The results from these experiments confirm that the performance gain of FINO is not attributable to the policy extraction scheme itself.
>
> &nbsp;
>
> ### **R3-4. Responses to Minor Questions**
> * Question about Figure 2
>
>     In Figure 2, we used a log scale.
>     We clarified this in the revised manuscript by updating the caption of Figure 2.
>
> * Question about $\\eta$
>
>     The value of $\\eta$ is fixed throughout the training process.
>     We updated Section 4.3 in the revised manuscript to make this explicit.

---

### Official Review · Reviewer_HJNR · 2025-11-01

**Soundness:** 3
**Presentation:** 3
**Contribution:** 3
**Rating:** 6
**Confidence:** 3

**Summary:**

This paper proposes FINO, a novel approach for offline-online RL. Centered on a Flow Matching generative model, it addresses the insufficient exploration capability and exploration-exploitation imbalance of existing methods through two key designs: in the offline pre-training phase, time-dependent Gaussian noise is injected to expand the coverage of the action space; in the online fine-tuning phase, an entropy-guided sampling mechanism is introduced, which dynamically adjusts temperature parameters based on policy entropy to balance exploration and exploitation. Across 45 complex scenarios from OGBench and D4RL, FINO achieves significantly better performance than baseline methods.

**Strengths:**

The paper exhibits a clear logical flow and high readability. The proposed algorithm is simple yet effective, supported by solid theoretical foundations and validated by striking experimental results.

**Weaknesses:**

Injecting noise helps enhance exploration capability, which is beneficial for the offline-to-online transition. However, why does this not compromise offline performance? After all, offline RL is inherently conservative and discourages exploration.

**Questions:**

1. Why $N_{sample}$ relates to the dimension of action space?
2. Do the baseline methods employ candidate action sampling? And would candidate action sampling significantly improve the performance of these baselines?

---

> ### Author Response · Authors · 2025-11-21
> **Authors' response to reviewer HJNR**
>
> We thank the reviewer for constructive feedback, which greatly contributed to strengthening the clarity and quality of our manuscript.
> Our responses to each point are provided below.
>
> &nbsp;
>
> ### **R2-1. Effect of Noise Injection on Offline Performance**
> The motivation behind our design is to train the policy over a broader action space so that its resulting diversity can be utilized for exploration.
> As discussed in Section 4.1, the noise injection in the flow matching objective increases the variance of the probability paths modeled by the flow model while keeping their mean unchanged.
> This ensures that the general behavior of the policy is preserved, which explains why its offline performance remains comparable to that of the baseline methods.
> We explicitly described this observation in Section 5 of the revised manuscript. (The current PDF at OpenReview is the revised paper file.)
>
> As the reviewer highlighted, offline RL is inherently conservative, and such conservatism can impede exploration.
> To counteract this, we incorporate an extrinsic sampling scheme based on action candidates.
> Even if the average behavior of the policy remains close to the dataset distribution, this sampling scheme allows the policy to consider diverse actions.
> When combined with the expanded action space induced by our noise injection, this mechanism enables the policy to perform more effective exploration during online fine-tuning.
>
> &nbsp;
>
> ### **R2-2. Regarding $N_\\text{sample}$**
> We thank the reviewer for highlighting this interesting point.
> Our method expands the action space that the policy learns by injecting noise into the flow matching objective.
> To effectively leverage this enlarged action space for exploration, we introduce a set of action candidates from which the agent selects an action.
> Since the volume of the space that requires exploration increases with the dimensionality of the environment’s action space, we design the number of action candidates to scale accordingly, ensuring that the method can fully utilize its increased diversity.
> We added an explanation of this design choice in Section 4.3 of the revised manuscript.
>
> To provide a clearer understanding of this hyperparameter, we included an experiment in Figure 7 of our revised manuscript that compares performance across different values of $N_\\text{sample}$.
> The results show that increasing the number of candidate actions improves performance, but this benefit saturates beyond a certain point.
> Because a larger $N_\\text{sample}$ also incurs higher inference cost, we choose a practical value that balances performance and computational efficiency.
>
> &nbsp;
>
> ### **R2-3. Action Candidate Sampling in Baselines**
> Our revised manuscript includes three baselines that also utilize candidate action sampling.
> * Action Noise baseline in Figure 4
>
>     In Figure 4, we compared our method with a simpler baseline that injects noise directly into the action.
>     This baseline employs the same candidate action sampling procedure as our method.
>
> * w/o Noise baseline in the left panel of Figure 6
>
>     Figure 6 presents a baseline where the proposed noise injection is removed from our method.
>     This baseline also adopts the candidate action sampling strategy.
>
> The results show that action candidate sampling alone is not sufficient to achieve strong performance without the support of our proposed noise injection mechanism.
>
> Moreover, in Table 1 and Figure 3, IFQL employs a much larger number of action candidates (approximately 3× to 10× more than FINO), yet it still underperforms compared to our method.
> This further supports that improvements of FINO are not attributed to candidate action sampling, but instead originate from the effective combination of noise injection and entropy-guided sampling.

---

### Official Review · Reviewer_czZj · 2025-11-01

**Soundness:** 3
**Presentation:** 3
**Contribution:** 3
**Rating:** 6
**Confidence:** 4

**Summary:**

This paper presents Flow Matching with Injected Noise for Offline-to-Online RL (FINO) that leverages flow matching policy for offline-to-online RL. It injects noise into policy training to facilitate exploration, which encourages action diversity. It balances exploration and exploitation by choosing the softmax of value functions over the action candidates. FINO is evaluated on a wide range of environments.

**Strengths:**

1. FINO explicitly integrates exploration into the learning process and balances between exploration and exploitation. The empirical result demonstrates superior performance over a wide range of tasks.
2. The paper is overall well-written and clear.

**Weaknesses:**

1. Does the noise injection step damage the performance in the offline learning stage?
2. Does FINO reduce policy randomness during evaluation?
3. The benefit of injecting noise during training (Eq. 7) rather than directly adding Gaussian noise to the output action is unclear.
4. The role of entropy guidance in improving performance is also not well explained. Does it enhance policy learning, or does it serve as a technique to improve test-time behavior?

**Questions:**

see weaknesses

---

> ### Author Response · Authors · 2025-11-21
> **Authors' response to reviewer czZj**
>
> We sincerely thank the reviewer for insightful and helpful comments.
> The feedback helps in enhancing both the clarity and the overall presentation of our work.
> We provide detailed responses to each point below.
>
> &nbsp;
>
> ### **R1-1. Offline Performance Under Noise Injection**
> FINO broadens the action space learned by the flow model through noise injection into the flow matching objective.
> However, as we clarified in Section 4.1, this noise injection increases the variance of the probability paths represented by the flow model without affecting their mean.
> Consequently, the general tendency of the policy remains consistent with the original one, and, as shown in Table 1 (offline scores indicated on the left side of each entry) and Figure 3 (performance at step 0), we observe no degradation in offline performance compared to our backbone model.
> Since maintaining offline performance is an important property of our approach, we clearly stated this in the revised manuscript. (The current pdf file at openview is the revised paper pdf file.)
>
> &nbsp;
>
> ### **R1-2. Regarding Randomness During Evaluation**
> To enable effective exploration during online fine-tuning, FINO learns a policy capable of generating diverse actions.
> However, directly evaluating such a diverse policy can lead to high variance in the measured performance, making the evaluation unreliable.
> To obtain a faithful and consistent assessment, we follow a similar procedure used during environment interaction, forming action candidates and selecting the one with the highest action-value according to the value function.
> Since the selected action is validated by the value function, this procedure provides a reliable evaluation.
>
> Moreover, considering the fact that the same evaluation procedure is applied to the baselines in Figure 4 (Action Noise) and Figure 6 (w/o Noise), the observed performance gains are due to the effectiveness of our noise injection and entropy guidance, rather than the advantage introduced by the evaluation procedure itself.
>
> &nbsp;
>
> ### **R1-3. Comparison with Direct Action Noise**
> We appreciate the reviewer for pointing out this important aspect.
> As the reviewer mentioned, adding Gaussian noise directly to the output actions can be considered a simple baseline, an alternative to our proposed noise injection strategy.
> However, the two approaches differ fundamentally in how they train the one-step policy that directly interacts with the environment.
>
> The one-step policy is optimized under the following objective:
>
> $$\\mathcal{L} _{\\pi} (\\omega) = \\mathbb{E}[- Q _{\\phi} (s, a _{\\omega} (s, z)) + \\alpha \\| a _{\\omega} (s, z) - a _{\theta} (s, z) \\|_2^2] $$
>
> The objective integrates a value-maximization term and a distillation term, meaning that the one-step policy is guided both toward high-value actions and toward the action distribution produced by the flow model.
> Since the flow model of FINO is trained over a wider action space, it provides guidance that allows the one-step policy to explore higher-value regions that lie beyond the support of the original dataset.
> This value-directed behavior, driven by the expanded flow model, supports more effective exploration, in contrast to approaches that simply add random noise to the actions.
>
> To verify the difference, we provided a clear experimental result (Figure 8) in Appendix D.2 of the revised manuscript.
> The result clearly shows that samples from FINO consistently move toward a higher-value region, whereas the baseline of simply adding noise to actions does not exhibit such value-directed behavior.
> We also included a performance comparison against this baseline in Section 6.1 of the revised paper, demonstrating that naive noise addition can even harm performance, further highlighting the effectiveness of our method.
>
> For clarity, we additionally introduced the explanation of the one-step policy training in Section 4.1 of the revised version.
>
> &nbsp;
>
> ### **R1-4. Regarding Entropy Guidance**
> During online fine-tuning, it is important to leverage exploration and exploitation effectively.
> Exploration allows the policy to discover new behaviors, while exploitation leverages these behaviors to improve performance.
> The proposed entropy guidance is introduced specifically to facilitate this balance during the learning process.
> By adjusting the sampling variable $\\xi$ accordingly to the policy’s entropy, it modulates the level of exploration and exploitation, allowing the policy to train efficiently within a limited online budget.
> Importantly, the entropy guidance is used only during training time. At inference time, the sampling variable is not applied, and thus the guidance does not influence test-time behavior.

---

### Author Response · Authors · 2025-11-21
**Common Response**

We sincerely thank the reviewers for their insightful comments.
The feedback greatly contributed to clarifying our intentions and enhancing the presentation of the manuscript.
We appreciate the time and effort dedicated to reviewing our submission.

The following list indicates the revisions made to our manuscript:
* **Figure 2**: We revised the caption to improve the clarity of the experimental results.
* **Section 4.1**: We added further explanation of how the proposed method influences the one-step policy to enhance the understanding of our approach.
* **Section 4.3**: We supplemented the description of hyperparameters to improve the clarity of the algorithm.
* **Section 5 (Results)**: We included clarification regarding the offline performance of the proposed method and expanded the discussion comparing it with IFQL.
* **Section 6.1 & Figure 4**: We introduced a new baseline, Action Noise, and included corresponding experiments.
* **Section 6.2 & Figure 5** (formerly Section 6.1 and Figure 4): We refined the explanation of this baseline to improve clarity.
* **Section 6.5 & Figure 7**: We added additional experiments analyzing the effect of $N_\text{sample}$.
* **Appendix D**: We included a comparison between the proposed noise injection scheme and a simple noise-injection alternative, along with results from a simple example.

We highlight all revised portions in blue in the updated manuscript available at OpenReview as PDF.

---

### Author Response · Authors · 2025-12-02
**Summary of Our Responses and Revisions during the Rebuttal Period**

Dear Area Chair,

We thank you for handling our submission in this difficult time.

In this paper, we proposed an efficient learning method for offline-to-online RL based on flow model.
Our method injects noise into the flow matching objective during training, enabling the flow model to cover a wide region of the action space, which facilitates effective exploration during online fine-tuning. Furthermore, we proposed entropy-guided sampling to balance exploitation and enhanced exploration during on-line fine tuning and evaluation. Our approach demonstrates state-of-the-art performance across diverse and challenging environments in the field of offline-to-online RL.

Overall, reviewers highly evaluated our work, and we summarize the major reviewer comments and our response/revision below:

**Comparison with Direct Noise Addition to Actions**

Reviewers czZj and QgFC questioned regarding the difference between our proposed noise injection and simply adding noise to the policy's action.
We provided a clear explanation on how our noise injection mechanism distinctively affects the learning of the one-step policy compared to simple action noise.
To clearly show the difference, we included additional analysis in Figure 8, which illustrates the distinct effect of our method.
We incorporated the detailed explanation into Section 4.1 and Appendix D of the revised paper.
Furthermore, we conducted a comparative experiment against the baseline utilizing simple action noise.
As shown in Figure 4, our approach enables significantly improved exploration.

**Impact of Action Candidate Sampling**

Reviewers HJNR and QgFC questioned whether the major performance gain may come from the action candidate sampling.
In response, we showed that our method achieves superior performance to the baselines equipped with action candidate sampling.
Based on this comparison, we clarified that action candidate sampling alone cannot account for the obtained significant performance gains.
Indeed, our noise injection into flow matching for enhanced exploration and entropy-guided action candidate sampling for balancing exploitation and enhanced exploration work together for performance improvement.

**Discussion on Offline Performance**

Reviewers czZj and HJNR commented on the potential impact of our method on offline performance.
In line with the theoretical explanation in Section 4.1, we clarified that our noise injection strategy does not alter the policy's underlying tendency, which explains why the offline performance remains stable.

We believe that our work makes a meaningful contribution and state-of-the-art performance in the field of offline-to-online RL, and the new experimental results, analysis, and revisions during the rebuttal period strengthened the completeness of our work.

We extend our gratitude to the reviewers for their constructive feedback and to the Area Chair for overseeing the review process.

Sincerely,

Authors

---

### Meta-Review · Area_Chair_MEXq · 2026-01-04

**Summary:**

The reviewers have common questions regarding direct noise addition to actions, and the impact of the noise injection on offline RL performance. From my point of view, all concerns have been addressed. After reading the paper (the paper is well-written), the reviewers' comments and the authors' rebuttal, I recommend acceptance.

**Reviewer Concerns:**

Reviewer czZj questioned the necessity and impact of the noise injection step during training, and asked whether FINO will reduce policy randomness and more explanation for entropy guidance. Reviewer HJNR also asked the impact of the noise injection on offline RL performance, and more explanation about the experiments and hyperparameters (specifically $N_{sample}$). Reviewer QgFC suggested that more explanation about the noise injection for flow matching is needed, and also raised questions regarding the baselines and ablations. From my point of view, all concerns have been addressed, and the reviewers have already provided relatively high scores.

**Reviewer Scores:**

I think that the reviewers may not change the score.

---

### Decision · Program_Chairs · 2026-01-26

Accept (Poster)